# RELATING TRANSFORMERS TO MODELS AND NEURAL REPRESENTATIONS OF THE HIPPOCAMPAL FORMATION

**James C.R. Whittington**[*]
University of Oxford & Stanford University

**Joseph Warren, Timothy E.J. Behrens**
University of Oxford & University College London

## ABSTRACT

Many deep neural network architectures loosely based on brain networks have recently been shown to replicate neural firing patterns observed in the brain. One of the most exciting and promising novel architectures, the Transformer neural network, was developed without the brain in mind. In this work, we show that transformers, when equipped with recurrent position encodings, replicate the precisely tuned spatial representations of the hippocampal formation; most notably place and grid cells. Furthermore, we show that this result is no surprise since it is closely related to current hippocampal models from neuroscience. We additionally show the transformer version offers dramatic performance gains over the neuroscience version. This work continues to bind computations of artificial and brain networks, offers a novel understanding of the hippocampal-cortical interaction, and suggests how wider cortical areas may perform complex tasks beyond current neuroscience models such as language comprehension.

## 1 INTRODUCTION

The last ten years have seen dramatic developments using deep neural networks, from computer vision (Krizhevsky et al., 2012) to natural language processing and beyond (Vaswani et al., 2017). During the same time, neuroscientists have used these tools to build models of the brain that explain neural recordings at a precision not seen before (Yamins et al., 2014; Banino et al., 2018; Whittington et al., 2020). For example, representations from convolutional neural networks (Lecun et al., 1998) predict neurons in visual and inferior temporal cortex (Yamins et al., 2014; Khaligh-Razavi & Kriegeskorte, 2014), representations from transformer neural networks (Vaswani et al., 2017) predict brain representations in language areas (Schrimpf et al., 2020), and lastly recurrent neural networks (Cueva & Wei, 2018; Banino et al., 2018; Sorscher et al., 2019) have been shown to recapitulate grid cells (Hafting et al., 2005) from medial entorhinal cortex. Being able to use models from machine learning to predict brain representations provides a deeper understanding into the mechanistic computations of the respective brain areas, and offers deeper insight into the nature of the models.

As well as using off-the-shelf machine learning models, neuroscience has developed bespoke deep learning models (mixing together recurrent networks with memory networks) that learn neural representations that mimic the exquisite *spatial* representations found in hippocampus and entorhinal cortex (Whittington et al., 2020; Uria et al., 2020), including grid cells (Hafting et al., 2005), band cells (Krupic et al., 2012), and place cells (O'Keefe & Dostrovsky, 1971). However, since these models are bespoke, it is not clear whether they, and by implication the hippocampal architecture, are capable of the general purpose computations of the kind studied in machine learning.

In this work we 1) show that transformers (with a little twist) recapitulate spatial representations found in the brain; 2) show a close mathematical relationship of this transformer to current hippocampal models from neuroscience (with a focus on Whittington et al. (2020) though the same is true for Uria et al. (2020)); 3) offer a novel take on the computational role of the hippocampus, and an instantiation of hippocampal indexing theory (Teyler & Rudy, 2007); 4) offer novel insights on the role of positional encodings in transformers. 5) discuss whether similar computational principles might apply to broader cognitive domains, such as language, either in the hippocampal formation or in neocortical circuits.

---

[*]Correspondence to: `jcrwhittington@gmail.com`

Note, we are not saying the brain is closely related to transformers because it learns the same neural representations, instead we are saying the relationship is close because we have shown a mathematical relationship between transformers and carefully formulated neuroscience models of the hippocampal formation. This relationship helps us get a better understanding of hippocampal models, it also suggests a new mechanism for place cells that would not be possible without this mathematical relationship, and finally it tells us something formal about position encodings in transformers.

## 2  TRANSFORMERS

Transformer Neural Networks (Vaswani et al., 2017) are highly successful machine learning algorithms. Originally developed for language, transformers perform well on other tasks that can be posed sequentially, such as mathematical understanding, logic problems (Brown et al., 2020), and image processing (Dosovitskiy et al., 2020).

Transformers accept a set of observations; $\mathbb{X} = \{\boldsymbol{x}_1, \boldsymbol{x}_2, \boldsymbol{x}_3, \cdots, \boldsymbol{x}_T\}$ ($\boldsymbol{x}_t$ could be a word embedding or image patch etc), and aim to predict missing elements of that set. The missing elements could be in the future, i.e. $\boldsymbol{x}_{t>T}$, or could be a missing part of a sentence or image, i.e. $\{\boldsymbol{x}_1 = \texttt{the}, \boldsymbol{x}_2 = \texttt{cat}, \boldsymbol{x}_3 = \texttt{sat}, \boldsymbol{x}_4 = ?, \boldsymbol{x}_5 = \texttt{the}, \boldsymbol{x}_6 = \texttt{mat}\}$.

**Self-attention.** The core mechanism of transformers is self-attention. Self-attention allows each element to 'attend' to all other elements, and update itself accordingly. In the example data-set above, the 4$^{\text{th}}$ element (?) could attend to the 2$^{\text{nd}}$ (cat), 3$^{\text{rd}}$ (sat), and 6$^{\text{th}}$ (mat) to understand it should be on. Formally, to attend to another element each element ($\boldsymbol{x}_t$ is a row vector) emits a query ($\boldsymbol{q}_t = \boldsymbol{x}_t \boldsymbol{W}_q$) and compares it to other elements keys ($\boldsymbol{k}_\tau = \boldsymbol{x}_t \boldsymbol{W}_k$). Each element is then updated using $\boldsymbol{y}_t = \sum_\tau \kappa(\boldsymbol{q}_t, \boldsymbol{k}_\tau) \boldsymbol{v}_\tau$, where $\kappa(\boldsymbol{q}_t, \boldsymbol{k}_\tau)$ is kernel describing the similarity of $\boldsymbol{q}_t$ to $\boldsymbol{k}_\tau$ and $\boldsymbol{v}_\tau$ is the value computed by each element $\boldsymbol{v}_\tau = \boldsymbol{x}_t \boldsymbol{W}_v$. Intuitively, the similarity measure $\kappa(\boldsymbol{q}_t, \boldsymbol{k}_\tau)$ places more emphasis on the elements that are relevant for prediction; in this example, the keys may contain information about whether the word is a noun, verb or adjective, while the query may 'ask' for any elements that are nouns or verbs - elements that match this criteria (large $\kappa(\boldsymbol{q}_t, \boldsymbol{k}_\tau)$, i.e. cat, sat, mat) are 'attended' to and therefore contribute more to the output $y_t$. Typically, the similarity measure is a softmax i.e. $\kappa(\boldsymbol{q}_t, \boldsymbol{k}_\tau) = \frac{e^{\beta \boldsymbol{q}_t \cdot \boldsymbol{k}_\tau}}{\sum_{\tau'} e^{\beta \boldsymbol{q}_t \cdot \boldsymbol{k}_{\tau'}}}$.

These equations can be succinctly expressed in matrix form, with all elements updated simultaneously:

$$\boldsymbol{y}_t = softmax(\frac{\boldsymbol{q}_t \boldsymbol{K}^T}{\sqrt{d_k}})\boldsymbol{V} \qquad \rightarrow \qquad \boldsymbol{Y} = softmax(\frac{\boldsymbol{Q}\boldsymbol{K}^T}{\sqrt{d_k}})\boldsymbol{V} \qquad (1)$$

Here $\boldsymbol{Q}$, $\boldsymbol{K}$, $\boldsymbol{V}$ are matrices with rows filled by $\boldsymbol{q}_t$, $\boldsymbol{k}_t$, $\boldsymbol{v}_t$ respectively, and the softmax is taken independently for each row. After this update, each $\boldsymbol{y}_t$ is then sent through a deep network ($f_\theta(\cdots)$) typically consisting of residual (He et al., 2016) and layer-normalisation (Ba et al., 2016b) layers to produce $\boldsymbol{z}_t = f_\theta(\boldsymbol{y}_t)$. $\boldsymbol{Z}$ is the output of the transformer which can then be used for prediction, or sent through subsequent transformer blocks.

**Position encodings.** Self-attention is permutation invariant and so tells you nothing about order of the inputs. Should the data be sequential (i.e. meaning depends on the order of elements, such as in language, or navigation as we will see later!), it is necessary to additionally encode the position/where $\boldsymbol{x}$ is in the sequence. This is typically done by adding a 'position encoding' that uniquely identifies each time-step ($\boldsymbol{e}_t$ - typically sines and cosines) to each input: $\boldsymbol{x}_t \leftarrow \boldsymbol{x}_t + \boldsymbol{e}_t$. Alternatively the position embedding can be appended i.e. $\boldsymbol{h}_t = [\boldsymbol{x}_t, \boldsymbol{e}_t]$, with self attention then performed using $\boldsymbol{h}_t$ as input.

## 3  TRANSFORMERS LEARN ENTORHINAL REPRESENTATIONS

Here we show that transformers (with a small modification) recapitulate spatial representations - grid and band cells - when trained on tasks that require abstract spatial knowledge.

**Spatial understanding task.** The task (more detail in Appendix) is to predict upcoming sensory observations $\boldsymbol{x}_{t+1}$ conditioned on taking an action $\boldsymbol{a}_t$ while moving around spatial environments (Figure 1a). For example, after seeing $\{(\boldsymbol{x}_1 = \texttt{cat}, \boldsymbol{a}_1 = \texttt{North}), (\boldsymbol{x}_2 = \texttt{dog}, \boldsymbol{a}_2 = \texttt{East}), (\boldsymbol{x}_3 = \texttt{frog}, \boldsymbol{a}_3 = \texttt{South}), (\boldsymbol{x}_4 = \texttt{pig}, \boldsymbol{a}_4 = \texttt{West}), (\boldsymbol{x}_5 = ?, \boldsymbol{a}_5 = \cdots)\}$, the aim is to predict

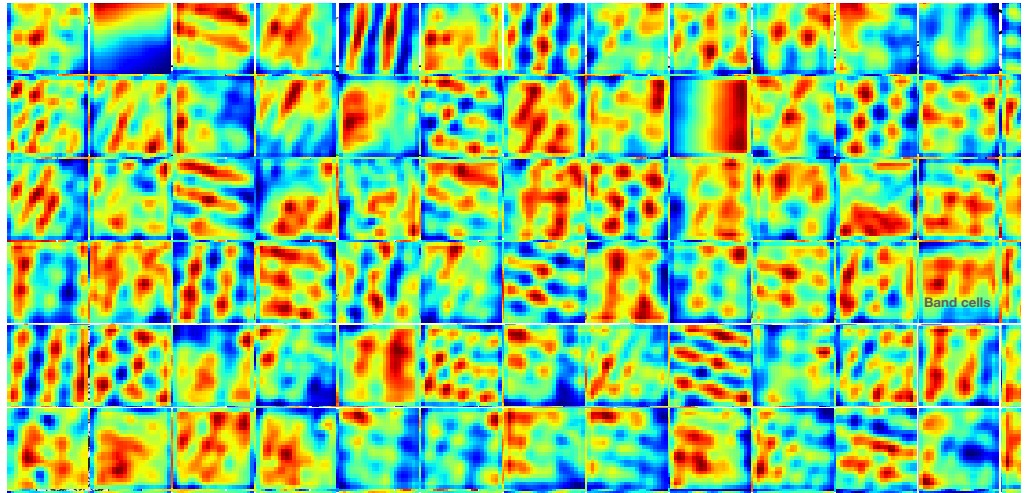

Figure 1: **(a)** Sequence prediction in spatial navigation tasks test abstract spatial understanding since some sensory predictions can only be done by knowing (generalising) certain rules e.g. `North + East + South + West = 0` or `Parent + Sibling + Niece = 0`. Note, we use sequences drawn from much larger graphs. **(b)** Transformer with recurrent position encodings. **(c)** Real grid cell rate-maps (Hafting et al., 2005). **(d-f)** Learned position embedding rate-maps (i.e. average activity at each spatial location; plots are spatially smoothed). **(d-e)** Resembling grid cells with **(e)** linear activation or **(e)** ReLu activation post transition. **(f)** Resembling band cells (Krupic et al., 2012).

$x_5 = $ `cat`. For simplicity, we treat sensory observations as one-hot vectors, thus the prediction problem is a classification problem.

When faced with an unseen stimulus-action pair (e.g. $x_4 = $ `pig`, $a_4 = $ `West` above; an action you have never taken at that stimulus before), successful prediction requires more than just remembering specific sequences of stimulus-action pairs; knowledge of the rules of space must be known; i.e. `North + East + South + West = 0` allows prediction of $x_5 = $ `cat`. Crucially, such rules *generalise* to any 2D spaces and may therefore be *transferred* to aid prediction in entirely novel 2D environments. This is powerful, since unobserved relations between observed stimuli can be inferred in a zero-shot manner.

However, these relational rules are not 'known' *a priori* and therefore must be learnt. We therefore train across multiple different spatial environments which share the same underlying 4-connected Euclidean structure (Figure 1a) - this means the model must learn and generalise the abstract structure of space to use for prediction in new environments.

To perform on these tasks, the three modifications to the transformer are:

1. Recall equation 1; $y_t = softmax(\frac{q_t K^T}{\sqrt{d_k}})V$, where $Q = HW_q$, $K = HW_k$, $V = HW_v$, and $H$ is a matrix of inputs and position encodings (i.e. its rows are $h_t = [x_t, e_t]$). We restrict these weight matrices such that queries ($Q$) and keys ($K$) are the same; $Q, K = EW_e$. We refer to this matrix as $\tilde{E}$. Thus the keys and queries only focus on *position* encodings. Meanwhile, values are exclusively dependent on the *stimulus* component of $H$ i.e. $V = XW_x$. We refer to this matrix as $\tilde{X}$.

$$y_t = softmax(\frac{q_t K^T}{\sqrt{d_k}})V \quad \rightarrow \quad y_t = softmax(\frac{\tilde{e}_t \tilde{E}^T}{\sqrt{d_k}})\tilde{X} \qquad (2)$$

   This is an extreme version of the realisation that, in transformers, best performance is when position encodings are used to compute keys and queries, but not values.

2. We use causal transformers; the key and value matrices contain the projected position encodings and sensory stimuli respectively at all *previous* time-steps (i.e. $e_{<t}$ and $x_{<t}$). This is equivalent to causal 'unmasking' as the agent wanders the environment accumu-

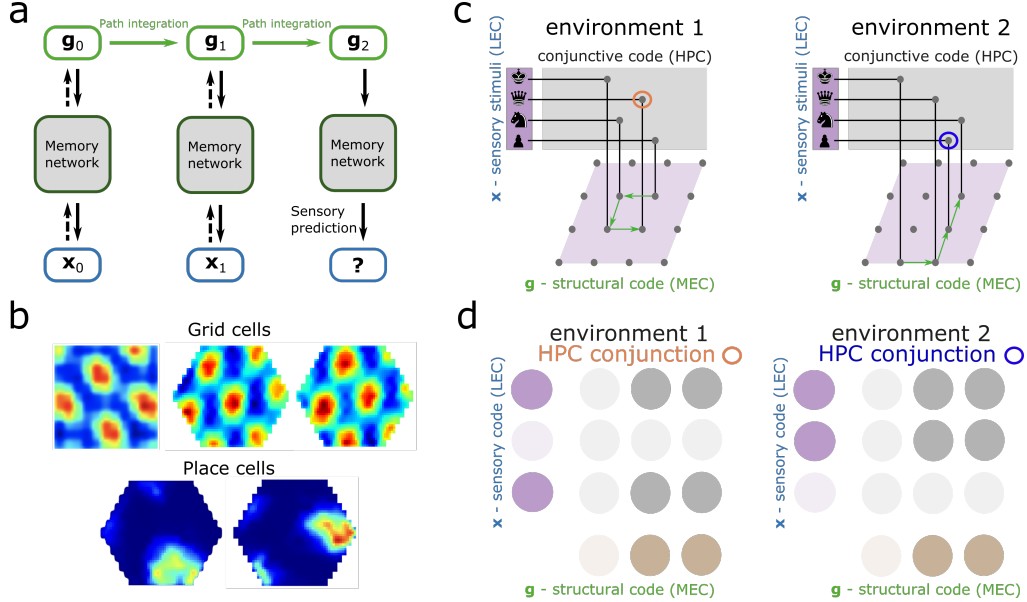

Figure 2: **(a)** The TEM model, with a path integration component (equation 3) and a memory network component (equation 5 and 6). TEM path integrates $g$ and makes sensory predictions $x$ via its memory network (dashed lines are additional connections for inference). **(b)** TEM recapitulates a host of empirically described cell representations (Whittington et al., 2020). Top/bottom row: example TEM MEC/Hippocampal representations (plots are spatially smoothed). Figures adapted from Whittington et al. (2020). **(c)** Schematic of TEM (adapted from Sanders et al. (2020)), showing that the *same* cortical representations (LEC and MEC) are reused in different environments allowing for generalisation, facilitated by *different* hippocampal combinations. **(d)** The TEM hippocampal conjunction is an outer product - cells receive input from particular MEC and LEC cells.

lating new experiences (not-yet-experienced stimulus-position pairs are inaccessible to the agent). Meanwhile the query at each time-point is the *present* positional encoding $e_t$.

3. The position encodings are recurrently generated (as in Wang et al. (2019); Liu et al. (2020)); $e_{t+1} = \sigma(e_t W_a)$, where $W_a$ is a *learnable* action-dependent weight matrix, and $\sigma(\cdots)$ is a non-linear activation function. This means that unlike traditional transformers, position encodings can be optimised and not the same for every sequence. It now becomes interesting to see what representations are learned.

These modifications are sufficient to learn spatial representations, in the position encodings, that mimic representations observed in the brain (Figure 1C; see Appendix for model and training details). The rest of this paper now explains why this is not a surprising result; namely we show that a transformer with recurrent positional encodings is closely related to current neuroscience models of the hippocampus and surrounding cortex (Whittington et al., 2020; Uria et al., 2020). Here we focus on the Tolman-Eichenbaum Machine (TEM) (Whittington et al., 2020), though the same principles apply for Uria et al. (2020).

The critical points are: 1) the memory component of TEM can be viewed as a transformer self-attention, since the TEM memory network is analogous to a Hopfield network (Hopfield, 1982) which have recently been shown to be closely related to transformers (Ramsauer et al., 2020); 2) TEM path integration (see below) can be viewed as a way to learn a position encoding.

## 4 TEM

The Tolman-Eichenbaum Machine (TEM; Figure 2, further details in Appendix) is a neuroscience model that captures many known neural phenomena in hippocampus (HPC) and entorhinal cortex

(medial/lateral; MEC/LEC). TEM is a sequence learner trained on tasks exactly like the one described in the previous section. TEM consists of two parts;

**1)** A module that aims to understand where it is in space, using a representation $\boldsymbol{g}$ to represent location. To update its location, TEM uses *path-integration* - the accumulation of self movement vectors $\boldsymbol{a}$ - enacted in a recurrent neural network:

$$\boldsymbol{g}_{t+1} = \sigma(\boldsymbol{g}_t \boldsymbol{W}_a) \tag{3}$$

Where $\boldsymbol{W}_a$ is a learnable action dependent weight matrix and $\sigma(\cdots)$ is a non-linear activation function. It is in this path-integrating representation$\boldsymbol{g}$ that TEM learns grid and other entorhinal cells for self-localisation (Figure 2b).

**2)** To make sensory predictions, location representations $\boldsymbol{g}$ alone are not enough; they must each link to a sensory observation $\boldsymbol{x}$, corresponding to the stimulus at that position. Note that these links are specific to an environment, since each environment consists of a different arrangement of stimuli in space (i.e. different stimulus-position pairings).

The linking is done by binding every element of $\boldsymbol{g}$ with every element of $\boldsymbol{x}$, in other words an outer product that is flattened back into a vector;

$$\boldsymbol{p} = flatten(\boldsymbol{x}^T \boldsymbol{g}) \tag{4}$$

These conjunctive $\boldsymbol{p}$ representations are stored in 'fast weights' [1]via Hebbian learning;

$$\boldsymbol{M}_t = \sum_{\tau=1}^{t-1} \boldsymbol{p}_\tau^T \boldsymbol{p}_\tau \tag{5}$$

And they can later be retrieved using an attractor network (a continuous version of the Hopfield network). Here a query vector $\boldsymbol{q}$ (details next paragraph) is inputted into the network and updated via;

$$\boldsymbol{q} \leftarrow \sigma(\boldsymbol{q}\boldsymbol{M}_t) \tag{6}$$

where $\sigma(\cdots)$ is a non-linear activation function; a ReLu in TEM. Crucially, because the memories are formed using both $\boldsymbol{g}$ and $\boldsymbol{x}$, they can be retrieved (pattern-completed) using just one of those representations alone i.e. 'what did I see the last time I was here' or 'where was I the last time I saw this'. To retrieve a memorised conjunction $\boldsymbol{p}$, TEM imagines (path-integrates) the next location $\boldsymbol{g}$ and provides this as input to the attractor network in the form $\boldsymbol{q} = flatten(\mathbb{1}^T \boldsymbol{g})$. Equation 6 is then iterated until a memory is retrieved.

Finally, to make sensory predictions, the retrieved conjunctive memory ($\boldsymbol{p}_t^{retrieved}$) is 'deconjunctified' into sensory and location components. The sensory component is obtained by unflattening $\boldsymbol{p}_t^{retrieved}$ and summing over the $\boldsymbol{g}$ dimension (Figure 8);

$$\boldsymbol{x}_t^{retrieved} = sum(unflatten(\boldsymbol{p}_t^{retrieved}), 1) \tag{7}$$

Finally, to make the sensory prediction $\boldsymbol{x}_t^{retrieved}$ is fed through a MLP $\boldsymbol{z}_t = f_\theta(\boldsymbol{x}_t^{retrieved}))$ to classify (predict) the upcoming sensory observation.

It is also possible, and often helpful, to project $\boldsymbol{g}$ and $\boldsymbol{x}$ via $\boldsymbol{W}_g$ and $\boldsymbol{W}_x$; $\tilde{\boldsymbol{g}} = \boldsymbol{g}\boldsymbol{W}_g$ and $\tilde{\boldsymbol{x}} = \boldsymbol{x}\boldsymbol{W}_x$ before they are combined conjunctively [2].

## 5   TEM AS A TRANSFORMER

Here we show that the above equations of TEM can be written so that: 1) the memory retrieval components looks like a transformer self-attention; 2) the path integration representation, $\boldsymbol{g}$ look like position encodings.

---

[1]We note such 'fast weights' have previously been thought of as an alternative to the LSTM (Ba et al., 2016a).

[2]In fact, in the TEM code online, $\boldsymbol{W}_g$ and $\boldsymbol{W}_x$ are set as fixed weight matrices, where $\boldsymbol{W}_g$ sub-samples $\boldsymbol{g}$ and $\boldsymbol{W}_x$ transforms (compresses) $\boldsymbol{x}$ from a one-hot to a two-hot representation.

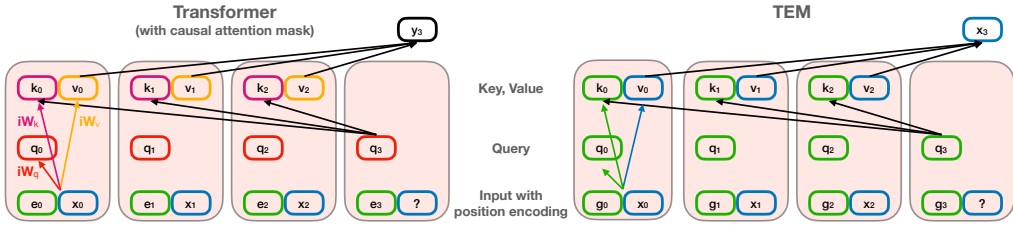

Figure 3: Self-attention in **(a)** Transformers and **(b)** TEM.

1) When considering the TEM memory retrieval process more closely (in this analysis, for direct comparison, we are only considering 1 attractor step in TEM with no non-linearity), we see that the attractor update $q_t M_t = q_t \sum_\tau^t p_\tau^T p_\tau$ is simply equal to

$$p_t^{retrieved} = \sum_\tau^t [q_t p_\tau^T] p_\tau \qquad (8)$$

Since $[q_t p_\tau^T]$ is just a dot-product ($[q_t \cdot p_\tau]$), a single step of the attractor just retrieves memories weighted by their similarity (dot product) to the query. As noted by Ramsauer et al. (2020), this is exactly like a transformer but without the softmax scaling the dot-products. Thus the TEM memory retrieval process behaves like transformer self-attention.

2) We can however go further since TEM's input to the transformer (i.e. the TEM memories) are special; they are learnable and built from an outer product between $\tilde{g}$ and $\tilde{x}$ ($p_\tau = flatten(\tilde{x}_\tau^T \tilde{g}_\tau)$ ), and these memories can be retrieved by a query based on $\tilde{g}$ or $\tilde{x}$ alone (e.g. $q_t = flatten(\mathbb{1}^T \tilde{g}_t)$). Together, these properties mean we can reduce the above dot product even further;

$$[q_t p_\tau^T] = \bar{\bar{x}}_\tau [\tilde{g}_t \cdot \tilde{g}_\tau] \qquad \rightarrow \qquad p_t^{retrieved} = \tilde{g}_t \tilde{G}^T \Lambda_x P \qquad (9)$$

Where $\bar{\bar{x}} = \sum_i (\tilde{x}_\tau)_i$ and $\Lambda_x$ is a diagonal matrix with elements $\bar{\bar{x}}_\tau$ (see Appendix for an alternative derivation using vector elements). Thus to retrieve a conjunctive $p$ memory, all that was necessary is weighting past $p$ representations via 'self-attention' of $\tilde{g}_t$ to past representations $\tilde{G}$.

To simplify this even further, we consider what happens when we 'deconjunctify' $p_t^{retrieved}$ to obtain the **sensory component of the memory**. Following the TEM procedure described above (Figure 8);

$$\tilde{x}_t^{retrieved} = sum(unflatten(p_t^{retrieved}), 1) = \sum_\tau^t \tilde{x}_\tau \bar{\bar{g}}_\tau \bar{\bar{x}}_\tau [\tilde{g}_t \cdot \tilde{g}_\tau] = \tilde{g}_t \tilde{G}^T \Lambda_g \Lambda_x \tilde{X} \qquad (10)$$

Where $\Lambda_g$ is a diagonal matrix with elements $\bar{\bar{g}}_\tau = \sum_i (\tilde{g}_\tau)_i$. Now all that is necessary to retrieve the sensory component of the memory is weighting past $\tilde{x}$ representations with via 'self-attention' of $\tilde{g}_t$ to past representations $\tilde{G}$. This equation is now very similar to equation 1 except without the softmax and with additional weightings $\Lambda_x$ and $\Lambda_g$. These weighting however are likely learned to be constant ($\alpha$) because otherwise some memories will be preferentially retrieved. In this case TEM is retrieving memories using

$$\tilde{x}_t^{retrieved} = (\alpha \tilde{g}_t \tilde{G}^T) \tilde{X} \qquad cf. \qquad softmax(\frac{\tilde{g}_t \tilde{G}^T}{\sqrt{d_k}}) \tilde{X} \qquad (11)$$

Which can be seen to be very closely related to the transformer equation (shown on the right), and diagrammatically shown in Figure 3. The model presented in this paper utilises the full transformer softmax rule.

**The TEM-transformer**. Thus the TEM-transformer (TEM-t; from Section 3) is this transformer that is directly analogous to TEM. Additional modelling details (analogous to modelling details in TEM) can be found in the Appendix. TEM-t offers dramatic performance improvements over the original TEM model (Figure 4; code will be released on publication). In particular, 1) Sample

**a** 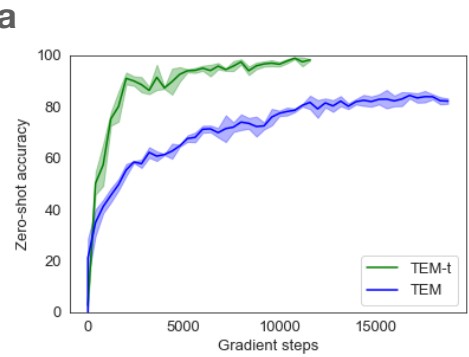

**b** 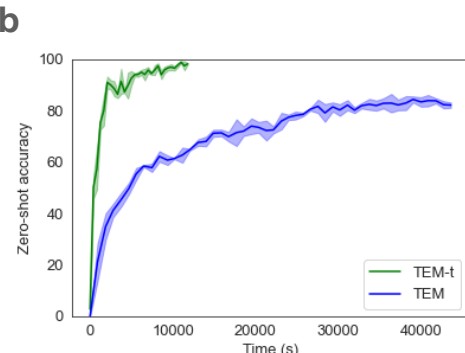

Figure 4: TEM-t is a more efficient learner than TEM, both in **(a)** sample efficiency and **(b)** time per gradient step. Zero-shot accuracy is prediction accuracy when taking links it has never taken before, but to a state it has visited before. Successful accuracy here is only possible with learned and generalised spatial knowledge. We have used the code from TEM from the TEM authors original code `https://github.com/djcrw/generalising-structural-knowledge`, and so have not optimised it for speed of learning etc, so we cannot claim this to be a fair comparison, nevertheless the difference is stark. We note that in the TEM paper, the authors say it takes up to 50,000 gradient updates for full training, whereas we stopped at 20,000.

efficiency is increased; TEM-t requires many fewer data samples than TEM, and thus training time is reduced 2) TEM-t can tackle much larger problems, with the ability to store and retrieve many more memories (not shown here). Additionally to improved performance, TEM-t learns grid cells (Figure 1) and has potential implications for what place cells are (see next section).

**Path integrating position encodings.** This leads us to an interesting observation; we see that TEM's representations for path integration $g$ plays the role of position encodings in transformers. However the structure of these positional encodings are not hard-coded, but instead *learned* via path integration (the structure of space!), with the particular position encoding depending on the particular sequence of actions taken. Other (non-spatial) structural representations could also be **learned** depending on the task at hand, i.e. grammar for language. This is a very different (and we think fruitful) re-understanding of position encodings; representing 'location' in a (learned) structure that can be inferred on the fly.

## 6  PLACE CELLS IN TRANSFORMERS

Here we discuss, and demonstrate, how TEM-t offers a new interpretation of place representations. To do so we utilise a recent suggestion of how the transformer update can be performed in biological hardware (Krotov & Hopfield, 2020). In particular, self-attention (equation 1) can be split into two steps which correspond to two pools of neurons (Figure 5A); 1) calculate $softmax(\frac{q_t K^T}{\sqrt{d_k}})$. 2) multiply by $V$. In this light, $K$ and $V$ can simply be seen as weight matrices between feature neuron (representing the query) and memory neurons (computing the softmax).

Since memory neurons are sparsely activated due to the softmax, they appear to have a spatial tuning for each environment resembling hippocampal place cells (Figure 5D-E; note Krotov & Hopfield (2020) stated memory neurons may correspond to place cells but without simulation). Similarly to experimentally recorded place cells, these neurons remap randomly between environments i.e. place cells being neighbours in one environment is not predictive of them being neighbours in another (unlike grid cells which maintain their phase neighbours across environments).

We can curate this architecture for the specifics of TEM-t. TEM-t explicitly considers factorised $g$ and $x$ representations (e.g. MEC and LEC), which project to feature neurons in hippocampus (or still in cortex). Thus the feature neurons consist of two separate sub-populations, $\tilde{g} = gW_g$ and $\tilde{x} = xW_x$, but which can connect to the same memory neurons in hippocampus (Figure 5B-C). These feature sub-populations are updated alternately rather than simultaneously, depending on the direction of retrieval; for example, when retrieving $\tilde{x}$ the $\tilde{g}$ feature neurons stay constant while the $\tilde{x}$

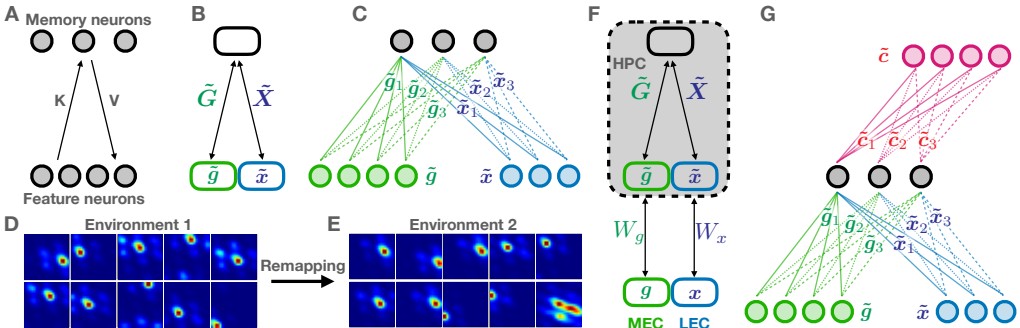

Figure 5: TEM-Transformer neural architecture. **(a)** Krotov & Hopfield (2020) describe a neurally plausible architectural instantiation the 'Hopfield networks is all you need' with a separation between 'feature' neurons (i.e. $h$) and memory neurons (i.e. softmax($q_t K^T$)). **(b-c)** This can be extended for TEM-t, but now the feature neurons are not all updated simultaneously, but only those across brain regions. **(d)** Memory neurons resemble hippocampal place cells and **(e)** remap randomly across environments. **(f)** A possible architecture where cortical neurons project to feature neurons in hippocampus which in turn project to memory neurons in hippocampus. **(g)** Additional brain regions can be included easily in this architecture with minimal increase in hippocampal neuron number.

neurons are updated (in turn updating $x$ in LEC). In this vein, hippocampal memories link together cortical representations in potentially disparate brain areas. Thus TEM-t instantiates hippocampal indexing theory (Teyler & Rudy, 2007), which states that hippocampus provides an index that binds together cortical patterns across different brain regions.

The randomly remapping place cells described one paragraph ago cannot be the full picture since we know that place cell remapping is not random; instead individual place cells preferentially remap to locations consistent particular grid cell firing (as predicted by conjunctive memory cells $p$ in TEM and verified experimentally in Whittington et al. (2020)). However another mechanism for this phenomena born from TEM-t could be as follows. Should the feature neurons exist in hippocampus (Figure 5F) then there will be hippocampal spatial cells $\tilde{g}$ that maintain their relationship to grid cells across different environment (as they are inherited from $g$ via a projection $\tilde{g} = g W_g$). Thus across the population of hippocampal cells, there will be those that maintain their relationship to grid cells (e.g. $\tilde{g}$ and those that don't (e.g. memory neurons and $\tilde{x}$), but the population effect will exist, just like what is experimentally observed.

As a note, the particular relationship of our model to the model of Krotov & Hopfield (2020), is what they refer to as a 'type B' model. These are models with contrastive normalisation on the memory neurons (via a softmax in our case), as opposed to 'type B' models which have a power activation function on the memory neurons. TEM (left hand side of Equation 11) corresponds to a linear activation function on the memory neurons, and is directly analogous to the original Hopfield energy. Secondly, the notion that there are two types of feature neurons that can be bound together in the same memory, was explored in Krotov & Hopfield (2016) where pixel intensities were associated with labels of those images. In TEM-t, one of the feature vectors, $g$, is learned via a RNN and structures itself according to the underlying task structure.

As an additional aside, we note that Krotov & Hopfield (2020) architectures does not solve the scaling problem of conventional Hopfield networks; the number of memories that original Hopfield networks could store scaled linearly with the dimensionality of the recurrent attractor network (Amit et al., 1985). While recent analytical work has shown with exponential power activation functions, the number of memories that can be stored to scale as $2^{\frac{N}{2}}$, where $N$ is the dimensionality of the feature neurons (Demircigil et al., 2017). This is a considerably more favourable scaling. However, unfortunately the architecture from Krotov & Hopfield (2020) instead requires a growing number of memory neurons (one for each memory), so the number of memories is still linear with the number of neurons! We note that mathematically derived scaling law was for an exponential activation function, not with a softmax as we use here.

# 7 DISCUSSION

We have shown that TEM, a current model of the hippocampal formation, is closely related to a transformer with recurrent position encodings. We now consider some wider implications for neuroscience.

**Multiple cortical inputs to hippocampus.** TEM considers hippocampal conjunctions between two cortical regions ($g$ and $x$). It is, however, possible to consider conjunctions of more than two brain regions. Indeed hippocampal neurons often respond to more than two task variables (McKenzie et al., 2014). In TEM, the naive approach of a 'triple' (or higher) conjunction would increase the number of hippocampal neurons would increase by a factor of $n_c$; the number of neurons from brain region $\tilde{c}$. TEM-t does not scale so badly. Instead it just requires an additional $n_c$ feature neurons, and the number of memory neurons can stay the same since the each hippocampal memory neuron can simply index a memory across three (or more), rather than two, brain regions (Figure 5G).

With multiple inputs to hippocampus $[\tilde{x}, \tilde{g}, \tilde{c}, \ldots]$, any subset of those brain areas can reinstate a memory in the other brain regions i.e. $\tilde{x}$ and $\tilde{g}$ can reinstate a $\tilde{c}$ memory or $\tilde{g}$ alone could reinstate $\tilde{x}$ and $\tilde{c}$ memories. As an analogy to the TEM triple conjunction, TEM-t proposes that $\tilde{c}_t$ is updated via $\tilde{c}_t \leftarrow softmax((\tilde{g}_t \tilde{G}^T) \odot (\tilde{x}_t \tilde{X}^T))\tilde{C}$, where $\odot$ is an element wise product. We note an alternate, and perhaps more intuitive, option could be $\tilde{c}_t = softmax(\tilde{g}_t \tilde{G}^T + \tilde{x}_t \tilde{X}^T)\tilde{C}$ - this is equivalent to two 'double' conjunctions in TEM, one between $\tilde{g}$ and $\tilde{c}$, the other between $\tilde{x}$ and $\tilde{c}$. This is important for TEM as it requires many fewer hippocampal neurons than a 'triple' conjunction.

**Beyond hippocampus: Cortex as a Transformer.** We have considered transformers as a model of hippocampus and its connections. We know, however, that transformer representations predict language areas (Schrimpf et al., 2020), and that patients can talk and comprehend just fine with major hippocampal deficits (Elward & Vargha-Khadem, 2018). This indicates that the transformer, and TEM-like models, may also model other brain regions, such as language areas, that are seemingly independent from hippocampus (related ideas discussed in Hawkins et al. (2019); Lewis (2021) but specifically for grid cells in neocortex). This raises two questions. Firstly what is the analogue of spatial positional encodings for higher order tasks such as language, and secondly what takes the role of the memory neurons if not hippocampus. We offer some thoughts in the following two paragraphs.

In spatial tasks, TEM and TEM-t learn positional encodings that mirror the structure of space. The implication is that positional encoding should reflect the abstract underlying properties of the task at hand. In language for example, this structure is grammar. This contrasts to the typical positional encodings in Transformers - sines and cosines - which represent a linear structure. It is our contention that positional encodings that are inferred on the fly and consist of previously learned structures (like the spatial case we have considered) would offer an interesting and potentially fruitful research direction in problems of language, maths, and logic.

If the transformer were solely instantiated in cortex, then what about the memory neurons? It is possible that the memory neuron equivalent exists in cortex too, but for these tasks, since it is not necessary to store long term memories or bind knowledge across multiple brain areas hippocampus is not required; so short term cortical memory neurons suffice.

# 8 CONCLUSION

We have shown that transformers with recurrent positional encodings reproduce neural representations found in rodent entorhinal cortex and hippocampus. We then showed these transformers are close mathematical cousins to models of hippocampus that neuroscientists have developed over the last few years. We hope this work brings neuroscience and machine learning closer together, and offers understanding for both sides; for neuroscientists a road map to understanding cortical areas beyond the hippocampal formation; for machine learners a greater understanding of positional encodings in transformers.

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

## A  APPENDIX

### A.1  THE MATHS USING ELEMENTS

Here we derive the main results using vector and matrix elements. Since each element in $\boldsymbol{p}$ is multiplicative combination of every pair of elements in $\tilde{\boldsymbol{x}}$ and $\tilde{\boldsymbol{g}}$, then

$$p_{ij} = \tilde{g}_i \tilde{x}_j \tag{12}$$

Where we label the elements of vector $\boldsymbol{p}$ with two indices even though it is a vector. Similarly, the memory matrix uses 4 indices;

$$M_{ijkl} = \sum_\tau \tilde{g}_i^\tau \tilde{x}_j^\tau \tilde{g}_k^\tau \tilde{x}_l^\tau \tag{13}$$

**Retrieving a memory via path integration.** Querying the network becomes $q_{ij} = \tilde{g}_i^{PI}$, which reduces to

$$\begin{aligned}
\sum_{ijk} q_{ij} M_{ijkl} &= \sum_\tau \sum_{ijk} \tilde{g}_i^{PI} \tilde{g}_i^\tau \tilde{x}_j^\tau \tilde{g}_k^\tau \tilde{x}_l^\tau \\
&= \sum_\tau \tilde{x}_l^\tau (\sum_k \tilde{g}_k^\tau)(\sum_i \tilde{g}_i^{PI} \tilde{g}_i^\tau)(\sum_j \tilde{x}_j^\tau) \\
&\rightarrow \tilde{\boldsymbol{g}}_t^{PI} \tilde{\boldsymbol{X}}^T \boldsymbol{\Lambda}_g \boldsymbol{\Lambda}_x \tilde{\boldsymbol{G}}
\end{aligned} \tag{14}$$

**Retrieving a memory using a sensory observation alone.** Similarly when the query is the sensory observation alone $q_{ij} = x_j$

$$\begin{aligned}
\sum_{ijl} q_{ij} M_{ijkl} &= \sum_{ijl\tau} \tilde{x}_j \tilde{g}_i^\tau \tilde{x}_j^\tau \tilde{g}_k^\tau \tilde{x}_l^\tau \\
&= \sum_\tau \tilde{g}_k^\tau (\sum_l \tilde{x}_l^\tau)(\sum_i \tilde{g}_i^\tau)(\sum_j \tilde{x}_j \tilde{x}_j^\tau) \\
&\rightarrow \tilde{\boldsymbol{x}}_t \tilde{\boldsymbol{X}}^T \boldsymbol{\Lambda}_g \boldsymbol{\Lambda}_x \tilde{\boldsymbol{G}}
\end{aligned} \tag{15}$$

**Retrieving a memory using a sensory observation and path integration.** When the query is $q_{ij} = g_i^{PI} x_j$, and we want to retrieve a position encoding memory

$$\begin{aligned}
\sum_{ijl} q_{ij} M_{ijkl} &= \sum_{ijl\tau} \tilde{g}_i^{PI} \tilde{x}_j \tilde{g}_i^\tau \tilde{x}_j^\tau \tilde{g}_k^\tau \tilde{x}_l^\tau \\
&= \sum_\tau \tilde{g}_k^\tau (\sum_l \tilde{x}_l^\tau)(\sum_i \tilde{g}_i^{PI} \tilde{g}_i^\tau)(\sum_j \tilde{x}_j \tilde{x}_j^\tau) \\
&\rightarrow (\tilde{\boldsymbol{x}}_t \tilde{\boldsymbol{X}}^T \odot \tilde{\boldsymbol{g}}_t^{PI} \tilde{\boldsymbol{G}}^T) \boldsymbol{\Lambda}_g \tilde{\boldsymbol{G}}
\end{aligned} \tag{16}$$

### A.2  DETAILS OF IMPLEMENTATION

Code will be made available at `https://github.com/djcrw/generalising-structural-knowledge`.

**Scaling beta.** Since the number of memories is not constant, we use an adaptive $\beta$ parameter in the softmax. This is because normalisation term in the softmax sums the number of memories, and so more memories down-weights probabilities. We want a self-attention value to not be affected by the number of elements in the set. In particular, we weight the softmax by $log(n_{memories})$.

**Losses.** We use same set of losses as in TEM;

- a one-step sensory prediction cross entropy loss, i.e. using $\boldsymbol{g}_t^{PI}$ as input to the memory retrieval process

- a sensory prediction cross entropy loss using $g_t$ as input to the memory retrieval process
- a squared error loss between $g_t$ and $g_t^{PI}$
- l2 weight regularisation
- l2 regularisation on $g_t$

**Normalisation.** We find that using layernorm (Ba et al., 2016b) on the positional encodings (not in the RNN, but on the input to transformer) to be beneficial, since the memory retrieval process can then be standardised - no one memory is up-weighted relative to others. Using layernorm before self-attention in transformers is common practice (Vaswani et al., 2017). For simplicity, we use fixed weights on the layer norm, i.e. is is just a z score of $g$, Since we use a one-hot encodings of our sensory representations, they are already normalised.

**Stabilising position representations.** Recurrently generated positional encodings accumulate noise and drift. While bespoke path integration networks from neuroscience mitigate noise by enforcing their neural representations to stay close to a neural manifold (Burak & Fiete, 2009), this can not be guaranteed in learned recurrent networks. One method of stabilisation is via sensory landmarks - i.e. 'what positional encoding did I have the last time I saw this landmark'. In this vein TEM uses the following query to the memory network; $q_t = flatten(\tilde{x}_t^T \mathbb{1})$, and **memories of positional encodings** are retrieved as;

$$\tilde{g}_t^{retrieved} = sum(unflatten(q_t M_t), 0) = \sum_{\tau}^{t} \tilde{g}_\tau \bar{\tilde{g}}_\tau \bar{\tilde{x}}_\tau [\tilde{x}_t \cdot \tilde{x}_\tau] = \tilde{x}_t \tilde{X}^T \Lambda_g \Lambda_x \tilde{G} \qquad (17)$$

The final position encoding, $g$ is computed on a basis of path integration ($g_t^{PI}$; equation 3) and stored landmark information ($g_t^{retrieved}$; equation 17), i.e. $g_t = g_t^{PI} + f_\theta(g_t^{retrieved}, g_t^{PI})(g_t^{retrieved} - g_t^{PI})$, where $g^{retrieved} = f_\theta(\tilde{g}^{retrieved})$ and $f_\theta(\cdots)$ are different MLPs.

We note that a better query to the memory network would be $q_t = flatten(\tilde{x}_t^T g_t^{PI})$ since sensory observations may be aliased, including $g_t^{PI}$ can help disambiguate such aliasing. In this case, the retrieved memory is $(\tilde{x}_t X^T \odot g_t^{PI} G) \Lambda_x G$. This is perhaps, different to what would be anticipated; using an element wise product rather than an addition. Translating this other TEM memory retrieval process into transformers we get $g_t^{retrieved} = f_\theta(softmax(\tilde{x}_t X^T \odot g_t^{PI} G) G)$, this can be thought of as another attention head i.e. from input $[g, x]$, the key and query 'attends' to $x$ while the value 'attends' to $g$.

**When to add memories.** Memories should not be added at every step, otherwise the self-attention mechanism would be biased towards memories (locations) that have been visited more frequently. This is not a problem typically addressed in transformer, but here to mitigate this issue memories are only added when existing memories for that particular conjunction do not already exist i.e. if there is a memory already with the present combination of $g$ and $x$ (similarity determined by dot product) then no new memory is added [3].

**Multiple iterations:** Though in our simulations we only used a single iteration of the transformer block, it is possible to use multiple iterations - just like a Hopfield network (as in Ramsauer et al. (2020)). Here the retrieved sensory memory $\tilde{x}_t^{retrieved}$ can be fed into the next iteration, along with positional encoding i.e. the path integrated $g$. In this case (see Appendix for derivation), TEM suggests that $\tilde{x}_t$ is iteratively updated via $\tilde{x}_t \leftarrow softmax(\tilde{x}_t \tilde{X}^T \odot \tilde{g}_t \tilde{G}) \tilde{X}$, but with the first iteration via $\tilde{x}_t \leftarrow softmax(\tilde{g}_t \tilde{G}) \tilde{X}$. The multiplicative term in the softmax is perhaps unexpected - it may be thought that it should be additive instead.

**Place-like representations without a softmax:** Here we chose a softmax activation function on the memory neurons to make the relationship between TEM and transformers exact. However was this choice necessary to observe place cells in the memory neurons? While we have not examined this experimentally, we suspect place-like representations would emerge for a variety of activations, since memories must still be sparsely activated for correct prediction. We leave this for future work to verify, but note again that power and exponential activations have been shown to have a

---

[3]The TEM model actually does something very much like this - memories are only added if they did not already exist - $M = \sum_{\tau}(p_\tau - \hat{p}_\tau)^T(p_\tau + \hat{p}_\tau)$, where $\hat{p}_\tau$ is the memory that was retrieved at $\tau$, i.e. only the un-predicted components of the memory $p_\tau$ are added.

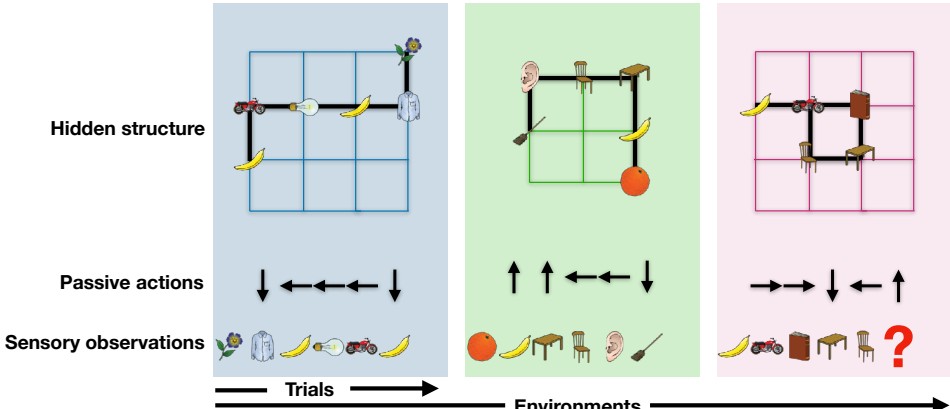

Figure 6: Learning to predict the next sensory observation in environments that share the same structure but differ in their sensory observations. TEM only sees the sensory observations and associated action taken, it is not told about the underlying structure - this must be learned.

much greater memory capacity (Krotov & Hopfield, 2016; Demircigil et al., 2017) than traditional Hopfield nets with linear activation.

### A.3 ADDITIONAL DETAILS OF THE SPATIAL TASK

We formalise a task type that not only relates to known hippocampal function, but also tests the learning and generalising of abstract structural knowledge. We formalise this via relational understanding on graph structures (a graph is a set of nodes that relate to each other).

Should one passively move on a graph (e.g. Figure 6), where each node is associated with a non-unique sensory observation (e.g. an image of a banana), then predicting the subsequent sensory observation tests whether you understand the graph structure you are in. For example, if you return to a previously visited node (Figure 6 pink) by a new direction - it is only possible to predict correctly if you know that a $right \rightarrow down \rightarrow left \rightarrow up$ means you're back in the same place. Knowledge of such loop closures is equivalent to understanding the structure of the graph.

We thus train our models on these graphs with it trying to predict the next sensory observation. Our models are trained on many environments sharing the same structure, e.g. 2D graphs (Figure 6), however the stimulus distribution is different (each vertex is randomly assigned a stimulus). Should it be able to learn and generalise this structural knowledge, then it should be able to enter new environments (structurally similar but with different stimulus distributions) and perform feats of loop closure on first presentation.

Formally, given data of the form $D = \{(\boldsymbol{x}^k_{\leq T}, \boldsymbol{a}^k_{\leq T})\}$ with $k \in \{1, \cdots, N\}$ (which environment it is in), where $\boldsymbol{x}_{\leq T}$ and $\boldsymbol{a}_{\leq T}$ are a sequence of sensory observations and associated actions/relations (Figure 6), $N$ is the number of environments in the dataset, and $T$ is the duration of time-steps in each environment, our model should maximise its probability of observing the sensory observations for each environment.

The sensory stimuli are chosen randomly, with replacement, at each node. We understand that this is not like the real world, where adjacent locations have sensory correlations - most notable in space (though names in a family tree will be less correlated). Sensory correlations help with sensory predictions, thus if we use environments with sensory correlations, we would not know what was causing the learned representations, sensory correlations, or transition structure. To answer this question cleanly, and to know that transition structure is the sole cause, we do not use environments with sensory correlations.

### A.4 ADDITIONAL DETAILS OF TEM

We present a more detailed model schematic of TEM in Figure 7. We see there are two components to TEM - a RNN for understanding position ($g$, in green top of Figure 7) that also indexes memories via 'queries' $q = W_g g$. A memory network that binds together $x$ and $g$, via an outer product (middle green in 7, with more detail in Figure 8A). When the memory network is queried (red in Figure 7), it undergoes attractor dynamics to retrieve the full memory. To make a sensory prediction, the retrieved memory is 'deconjunctified' into a sensory representation (Figure 8C).

**Model flow.** TEM transitions through time and infers $g_t$ and $p_t$ at each time-step. $g_t$ is inferred before forming each new memory $p_t$. In other words variables $g$ and $p$ are inferred in the following order $g_t, p_t, g_{t+1}, p_{t+1}, g_{t+2} \cdots$. This flow of information is shown in a schematic in Figure 7.

Independently, at each time-step, the model model asks 'are the inferred variables what I would have predicted given my current understanding of the world (weights)'. I.e. 1) Is the inferred $g_t$ the one I would have predicted from $g_{t-1}$. 2) Is the inferred $p_t$ the one I would have predicted from $g_t$. 3) Is $x_t$ what I would have predicted from $p_t$. This leads to errors (at each timestep) between inferred and predicted variables $g_t$ and $p_t$, and between sensory data $x_t$ and its prediction.

At then end of a sequence, these errors are accumulated, with model parameters updated along the gradient (from back-propagation through time) that matches each others variables and also matches the data.

Since the model runs along uninterrupted, it's activity at one time-step influence those at later time-steps. Thus when learning (using back-propagation through time - BPTT), gradient information flows backwards in time. This is important as, should a bad memory be formed at one-time step, it will have consequences for later predictions - thus BPTT allows us to learn how to form memories and latent representations such that they will be useful many steps into the future.

### A.5 RELATED WORK

Here we discuss and compare other models that recapitulate spatial representations found in the brain. These models fall into several categories. 1) Auto-encoder like models trained on spatial representations (Dordek et al., 2016). 2) Graph representation models (Gustafson & Daw, 2011; Stachenfeld et al., 2017). 3) Latent state inference models (George et al., 2021). 4) Path integrating models with RNNs trained on spatial representations (Cueva & Wei, 2018; Banino et al., 2018). 5) Models mixing RNNs and memory networks that are trained on sequences of sensory observations.

We note that models (1-4) are learned using curated spatial representation, i.e. the modeller has already worked how how space works, and the model is finding an alternate representation of that space. This is not how you or I learn. Instead, we learn by extracting regularities from the sensory world, and transfer/generalise this knowledge from domain to domain. Model category (5) does unsupervised learning by predicting sequences of sensory observations alone. From just sensory observations, it can slowly build up a picture of how space is structured, and then transfer this knowledge to situations that share the same underlying spatial structure. Our model TEM-t fall in the model (5) category, along with TEM (Whittington et al., 2020) and other similar models Uria et al. (2020).

For model to make sensory predictions as fast as possible, it is necessary to have two components. a) A RNN that understand position, and b) a memory network that can remember what you see and where you see them. Now the model can be asked what 'what will you see when heading `East`' and correctly respond `Cat` even it it has never gone East at that location before. This is because it can simulate going East via its RNN and then retrieve the memory it had stored at that location (it must have visited the Cat location before thought!). The other models, e.g. an autoencoder, could not solve this task since it learns slow statistical structures between sensory observations over many many training examples. Thus it would be clueless to the entirely new configuration of sensory observations in the new environment. In sum, **it is necessary to have an understanding of position, and the ability to make and retrieve memories to successfully make sensory predictions as fast as possible**.

We describe the relationship between TEM (and thus the hippocampal-entorhinal system) as close **not** because it produces the same representations, instead we are saying the relationship is close

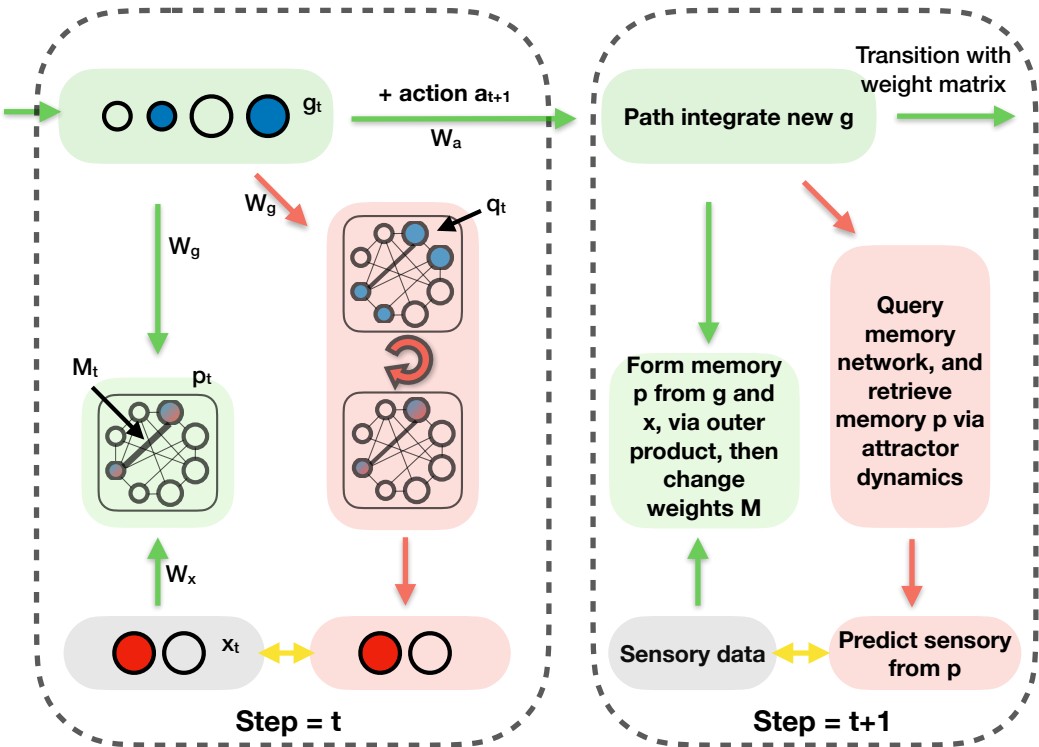

Figure 7: Schematic to show the model flow. Depiction of TEM at two time-points, with each time-point described at a different level of detail. Timepoint $t$ shows network implementation, $t+1$ describes each computation in words. Red is for model predictions, green is for updating model variables. We do not show the stabilising position encodings module here. Circles depict neurons (blue is $g$, red is $x$, blue/red is $p$); shaded boxes depict computation steps; arrows show learnable weights; looped arrows describe recurrent attractor. Black lines between neurons in attractor describe Hebbian weights $M$. $W_a$ are learnable, action dependent, transition weights. $W_g$ and $W_x$ are learnable projection matrices. Yellow arrows show training errors.

because **we have shown a mathematical relationship between the two models**. This could not happen for most ML models as most ML systems do not have both RNN structured representations and a memory system and so is not mathematically relatable to current models of the hippocampal formation. However, equipped with the mathematical relationship, we can say that the memory part of the transformer is related to the hippocampal memory system in TEM. And the positional encodings are related to the entorhinal grid RNN system in TEM.

## A.6 FURTHER LEARNED CELL REPRESENTATIONS

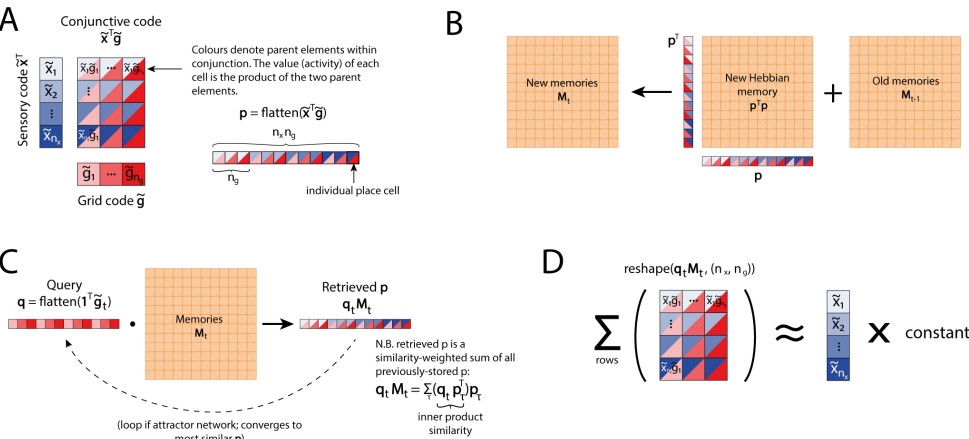

Figure 8: Memory formation and retrieval in TEM. **(a-b)** Memory formation. **(a)** Projected sensory sensory code $\tilde{x}$ and projected grid code $\tilde{g}$ are combined via an outer-product $\tilde{x}^T\tilde{g}$, which is flattened to obtain a vector of place cells $p$. Each place cell (denoted by a single diagonally divided cell) is a conjunction of an element from each of $\tilde{x}$ and $\tilde{g}$ (denoted by the two colours composing each cell). The activity of the place cell is the product of the values of these elements. **(b)** A new Hebbian memory $p^T p$ is added to the existing memory matrix $M$. **(c-d)** Memory retrieval. **(c)** Multiplication of the query $q$ with the memory matrix $M$ retrieves a place code $p$. This retrieved code is the sum of previously experienced codes, weighted by their similarity to the present query. This may be repeated iteratively to converge to the stored $p$ that is most similar to $q$. **(d)** The retrieved place code $p$ is reshaped and summed along the rows to average-out the $g$ components. The result is $\bar{\tilde{g}}\tilde{x}$.

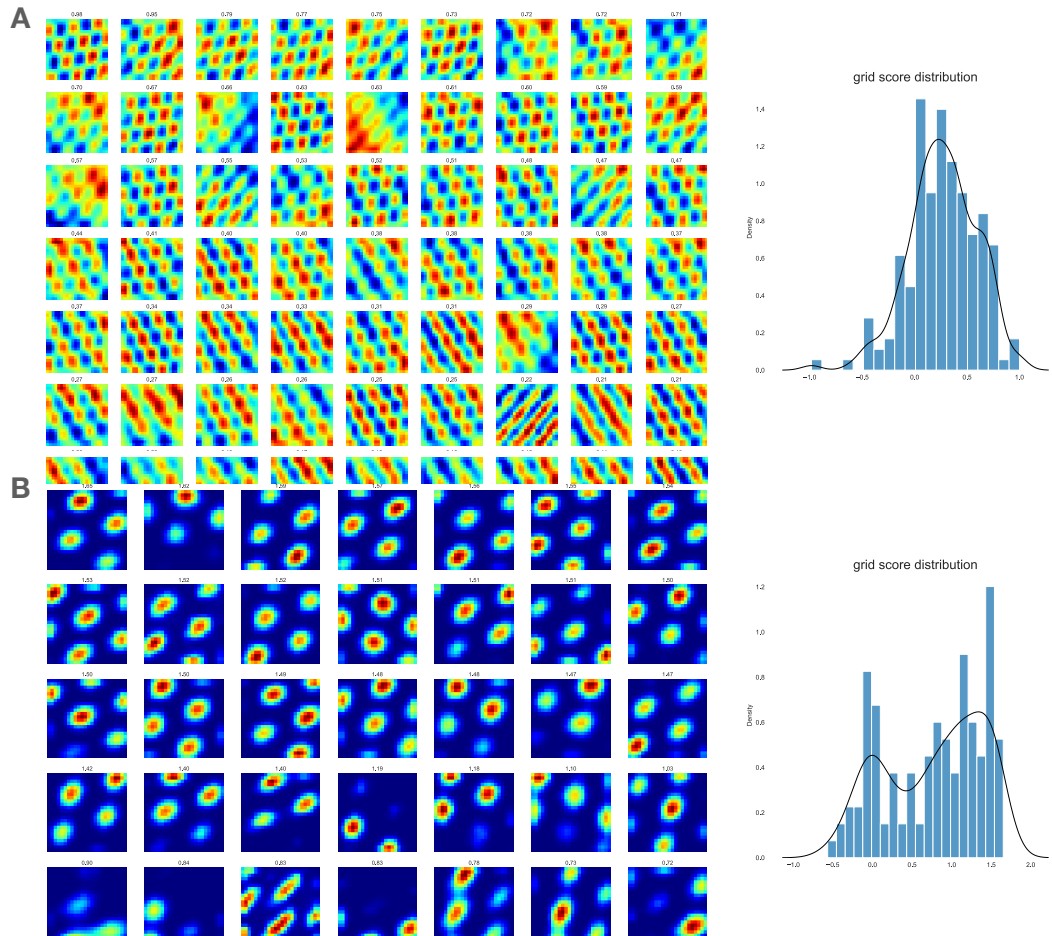

Figure 9: Learned grid cells ordered by grid score. We only show cells that are both active and have a grid score above 0. A gird score of 0.3-0.5 is generally considered to be a grid cell. The panels on the right hand side are the show the grid scores for the **whole population** of cells (though in some cases the grid score was not calculable e.g. if the cell has no activity; these cells are ignored in the analysis). **(a)** Grid cells learned with a linear activation function. **(a)** Grid cells learned with a ReLu activation function.

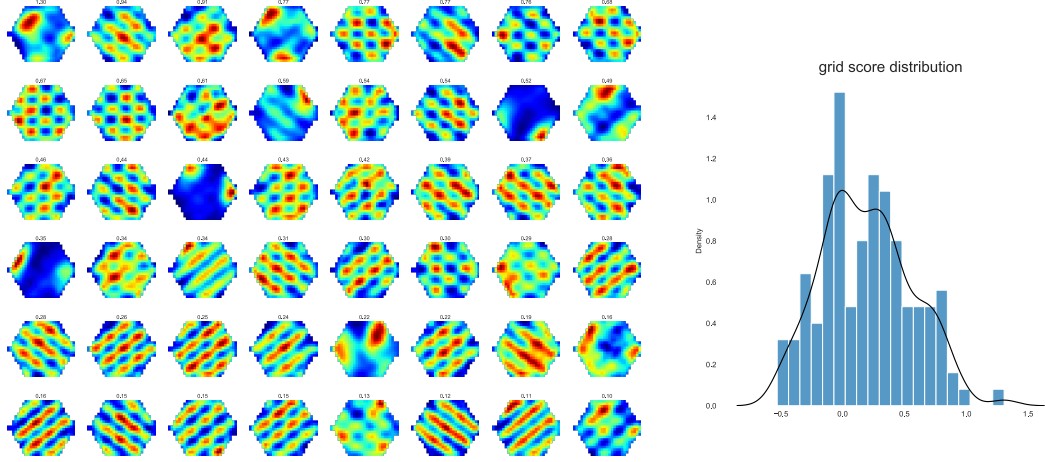

Figure 10: Learned grid cells in hexagonal 6-connected world.

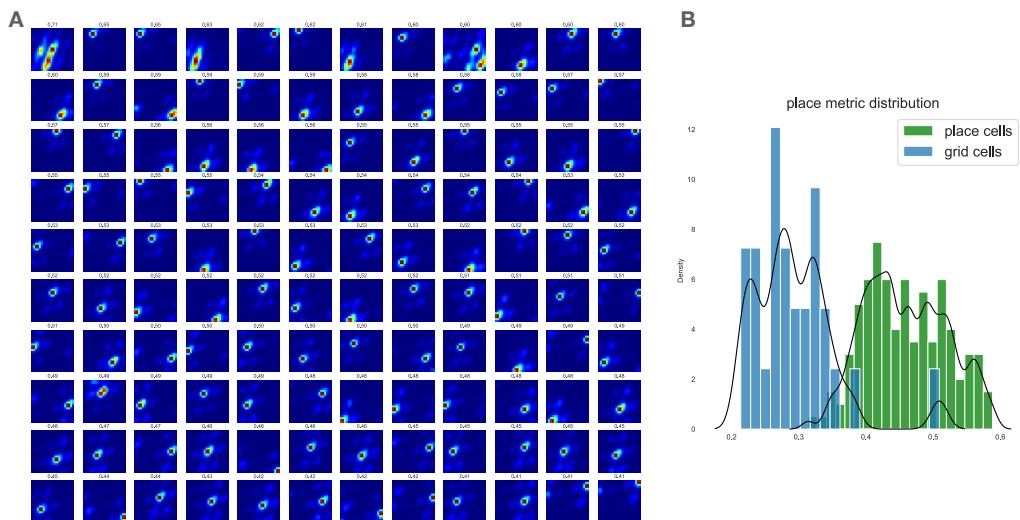

Figure 11: Place cells ordered by novel place cell metric. This metric assess how place-like the firing of each cell is. In particular, we look at the connected components of the firing rate map, and our metric is the ratio of 'firing mass' in the largest connected component versus all connected components. This metric is 1 if all the firing is in a single component, and it is lower if the firing is spread between components. **(a)** TEM-t learned memory place cells. **(b)** Our novel metric distinguishes between TEM-t RNN neurons (grid cells), and TEM-t memory neurons (place cells).

