# OpenReview forum: "Relating transformers to models and neural representations of the hippocampal formation"
_ICLR.cc/2022/Conference — ICLR 2022 Poster_

### Official Review · Reviewer_skR7 · 2021-10-30

**Correctness:** 4
**Technical Novelty And Significance:** 3
**Empirical Novelty And Significance:** 3
**Recommendation:** 8
**Confidence:** 3

**Main Review:**

*Strengths*

The authors show the full analogy between the Transformer architecture and the TEMs – unlike the existing literature which has only identified some similarities between the two (e.g. the similarity between the attractor update rule in TEM and the matrix product in self-attention).

The Transformer architecture proposed in this paper inherits the biological plausibility of TEMs and, at the same time, is considerably more sample-efficient, allowing for building computationally efficient models of the hippocampus and entorhinal cortex.

The proposed architecture is expected to scale well with an increase in the number of cortical inputs to the model thus potentially enabling the studies of multisensory integration.

The paper has a nice introduction to the Transformer architecture.

*Weaknesses*

Whereas the authors, due to their results, expect the Transformer architecture to inform the future models of the hippocampus – and the other way around – such work has not been done in the scope of the current paper.

I think it would be nice to include a slightly more detailed description of the TEM model in the paper. I like the equation-based description, offered here, better than the text-based one in the original TEM paper; however, this approach would benefit from defining the variables and, perhaps, briefly commenting on their role before using them (e.g. the q-variable)


**Summary Of The Paper:**

The authors show that the Transformer architecture, with a specific choice of parameters, is analogous to the Tolman-Eichenbaum Machine (TEM), a neuroscientific model successful at explaining phenomena related to grid cells in the entorhinal cortex, place cells in the hippocampus, and their coordinated remapping in new environments. The authors show that, compared to TEM, its Transformer counterpart is more sample-efficient and should better generalize to multisensory inputs, thus offering a computationally efficient basis for models of the hippocampus and entorhinal cortex.

**Summary Of The Review:**

The paper proves a parallel between the Transformers and the Tolman-Eichenbaum Machines, thus offering a computationally efficient and scalable implementation for this biologically-grounded model of the hippocampus. Even though no new biological insights were obtained in the scope of this paper using that knowledge, the parallel between the two frameworks deems useful for models of the hippocampus in follow-up works and highly relevant to the community. I, therefore, recommend accepting the paper.

---

> ### Author Response · Authors · 2021-11-22
> **Response to Reviewer skR7**
>
> Many thanks for your review! We are very grateful for your kind apprecation of our work, and your critiscims have been taken onboard and have greatly impreoved the manuscript. We hope that we have addressed your comments in our response below.
>
> **REVIEWER COMMENT** “Whereas the authors, due to their results, expect the Transformer architecture to inform the future models of the hippocampus – and the other way around – such work has not been done in the scope of the current paper.”
>
> We understand the reviewer’s concern (and disappointment) that models of neuroscience are not so easy to translate into models of ML. Recent history has largely been in the direction of flow from ML to neuroscience but not the other way round. This is a difficult problem to address!
>
> We firmly believe that drawing formal parallels between ML models and Neuroscience models is profitable for both parties - it is what lets us understand systems better and reveal new mechanisms. We have shown:
> 1) For neuroscientists: This mathematical relationship we show helps us get a better understanding of hippocampal models. It also suggests a new mechanism for place cells that would not be possible without this mathematical relationship. It also sheds new light on the hippocampal role in complex sequence learning e.g. the hippocampal role in language has that seemed at odds with the traditional hippocampal role in spatial cognition and memory, but now can be unified by its relationship to transformers.
> 2) For ML researchers: We have uncovered new insights into position encodings in transformers. Before they were just thought of as conveying position in a line via sines and cosines. This was done to achieve better empirical results. Now we can understand them as reflecting the underlying structure of the problem (e.g. physical space in our case). This means that not every sequence should be given the same position encoding - as different sequences can take different trajectories, and so be at different positions.
>
> **REVIEWER COMMENT** “I think it would be nice to include a slightly more detailed description of the TEM model in the paper. I like the equation-based description, offered here, better than the text-based one in the original TEM paper; however, this approach would benefit from defining the variables and, perhaps, briefly commenting on their role before using them (e.g. the q-variable)”
>
> Many thanks for this comment! It is clear to us now that we did not include enough explanations of the models - in particular of TEM.
>
> Firstly, we have clarified the q-variable in the main text, prior to its mention in the TEM section. We have also included a more detailed description of TEM in the appendix, along with companion figures. We do not put the text here as it is several paragraphs, but please see the paper for the text!

---

> > ### Comment · Reviewer_skR7 · 2021-11-22
> > **Thank you for your response**
> >
> > Dear Authors,
> >
> > Thank you for your response.
> >
> > The clarifications here and the changes made to the text do address my comments. I also acknowledge the significant effort to address the other Reviewers' comments which, in my opinion, have been reasonably addressed. That includes the Reviewer hKkg's comments about the claims of the paper which, I believe, were fully addressed.

---

> > > ### Author Response · Authors · 2021-11-22
> > > **Many thanks!**
> > >
> > > Many thanks for the quick reply - we really appreciate your comments!

---

> ### Comment · Reviewer_skR7 · 2021-11-28
> **After-rebuttal comments**
>
> I think the authors have done a good job at timely addressing the remaining concerns during the author-reviewer interaction so I'm raising my score. Thanks, everyone.

---

> > ### Author Response · Authors · 2021-11-28
> > **Many thanks!**
> >
> > Many thanks - we're really very grateful for this! Many thanks again for your initlal review - the paper is much clearer and better for it!

---

### Official Review · Reviewer_hKkg · 2021-10-31

**Correctness:** 3
**Technical Novelty And Significance:** 3
**Empirical Novelty And Significance:** Not applicable
**Recommendation:** 6
**Confidence:** 3

**Main Review:**

In general I am very curious and supportive of the effort to find similarities between the methods and representations that biological and artificial learning systems discover during their learning process. However we must be very strict with ourselves when we declare that such a similarity has been found:
  * (a) Is it sufficiently similar to be non-trivial?
  * (b) Can the emergence of this similarity be explained by other factors, such as experimental setup?
  * (c) Does the similarity occur only when we are using one particular ML algorithm or would it also emerge when attempted using other learning methods / algorithms / architectures that are appropriately configured?

The emergent patterns resembling grid cell structure that are shown to emerge in this paper closely resemble the patterns that emerged in an RL agent in this work: https://deepmind.com/blog/article/grid-cells. In that work Transformers were not used, but the pattern has emerged nevertheless. This begs the question whether Transformers are indeed *the* architecture that encodes spatial representation closest to hippocampus, or whether any ML model, properly incentivized, will form such representations?

Authors mention in the abstract that "this result is no surprise since it is closely related to current hippocampal models from neuroscience" about the claim "transformers, when equipped with recurrent position encodings, replicate the precisely tuned spatial representations of the hippocampal formation". Could we have written the same paper by just replacing Transformers with something else that could mimic neuroscientific model of hippocampal grid cells? If yes, then Transformers have no special significance here and should not enter the discussion.

I am looking forward to the discussion and am very to open to changing my rating upon further exploration of these matters.


Major concerns / criticisms / discussion
----------------------------------------

* Seeing how Transformers are very flexible models with high capacity, wouldn't it be the case that (with a little twist) one could fit them to become good representations of almost any phenomena? The theory that is put forward in this paper is that Transformers model representations that are close to those used by hippocampal formation. What kind of result would invalidate such a theory? Let's say that instead of Transformers we would use a black box method "X" that would tune its parameters in such a way, that they correlate well with hippocampal responses. Would that mean that "X" is good model of representations that are being used by hippocampus? Note that the more flexible "X" is and the more time and computational resources we have, the closer we can bring "X"'s activations to correlate with hippocampal responses. What would be a hypothetical empirical indication from your experiments that would lead you to conclude that "nah, Transformers are nothing like hippocampus"?

* Our brain did not learn in an environment with a 4-connected Eucledian structure. Wouldn't having this structure most assuredly lead to a representation that reflects the pattern of the structure? Let's say you would conduct the same experiment in an environment with a some kind of 10-connected star-shaped Eucledian structure -- wouldn't the patterns of representations formed by the Transformer would try to capture that star-shape?


Questions
---------

* page 3: "It now becomes interesting to see what representations are learned" -- taking into account the properties of the task and its spatial structure, what would we expect the representation that captures that structure to look like? Does it have to be a Transformer to capture that representation or could be captured, for example, by an autoencoder trained to find an efficient representation of the samples constituting the task?

* page 4: "Hopfield network (Hopfield, 1982) which have recently been shown to be closely related to transformers" -- Could you please elaborate what is meant by being related and how this relationship pertains to the claim that "a transformer with recurrent positional encodings is closely related to current neuroscience models of the hippocampus and surrounding cortex"?



Minor remarks
-------------
page 2: typo "Crucially, such rules generalises"

**Summary Of The Paper:**

The paper postulates a task that requires the ML model to capture the spatial nature of the task in order to perform well. The chosen ML method is a Transformer with positional encoder, where the encoding is not fixed, but is learnable, making it possible for the transformer to "choose" what kind of representation would it "prefer" for the task at hand. The Transformer is modified to act comparably to a neuroscientific model of hippocampal function (TEM). After the learning process the authors discover, that, when visualized, the representation that has emerged resembles the grid-cell pattern similar to the one that is being encoded by hippocampal place and grid cells. From there the works suggests the existence of a special relationship between the Transformer architecture and the mechanism by which hippocampus encodes spatial information.

**Summary Of The Review:**

In my current evaluation (prior to discussion with the authors) it seems to me that the claim that Transformers have a special kind of relation to hippocampal grid cells is not supported by the evidence. The evidence does show that the condition is "sufficient" (Transformers do form grid-like representations), but it does not show that it is also "necessary" (meaning it does not have to be a Transformer to form a grid-like representation). If my understanding of the evidence is correct, then it does not support the claim of the paper and I would recommend rejecting this result.In my current evaluation (prior to discussion with the authors) it seems to me that the claim that Transformers have a special kind of relation to hippocampal grid cells is not supported by the evidence. The evidence does show that the condition is "sufficient" (Transformers do form grid-like representations), but it does not show that it is also "necessary" (meaning it does not have to be a Transformer to form a grid-like representation). If my understanding of the evidence is correct, then it does not support the claim of the paper and I would recommend rejecting this result.

---

> ### Author Response · Authors · 2021-11-22
> **Response to Reviewer hKkg (1/4)**
>
> We thank the reviewer for their thoughtful comments - they have helped us clarify our arguments and have greatly improved this paper - many thanks! We respond to your points below, though beforehand we include a summary of our points. We hope that it is now clear what our contributions are and that these contributions are novel and worth publishing.
>
> To summarise our response:
>
> We apologise for our lack of clarity on the main results of this paper. It was not our intention to just claim that transformers with their high capacity can solve many problems therefore transformers could potentially model any brain region. Instead, our main claims are that
> 1) A transformer with recurrent position encodings learns grid cell representations in its RNN, and place cell representations in its softmax over memories.
> 2) A transformer with recurrent position encodings is **precisely relatable via mathematics** to current neuroscience models of the hippocampal formation. Note, we are not saying the TEM/ the brain is closely related to transformers because it produces the same representations, we are saying the relationship is close because we have shown a **mathematical relationship between the two models**. It is not possible to make mathematical relations between most models! Least of all between AI models and neuroscience models.
> 3) This relationship helps us get a better understanding of hippocampal models. It also suggests a new mechanism for place cells that would not be possible without this mathematical relationship.
> 4) It also tells us something formal about position encodings in transformers, beyond simple linear structure (sines and cosines) which had always just been used for their empirical success. Here we say that position encodings must reflect the underlying structure of the problem (e.g. physical space in our case).
>
> In general, it is not immediately clear that neuroscience models should be similar to models from ML. Mechanistic models from neuroscience models arise from carefully thinking about specific brain regions, the type of problems they are trying to solve, the types of cells that have been recorded in these regions, and anatomical connections between brain regions. While some ML models (e.g. CNNs) have been inspired from brain processing, transformers were not, thus it is interesting that they are in fact closely related to independently developed models of the hippocampal formation (that were also developed without transformers in mind!).
>
> We firmly believe that drawing formal parallels between ML models and Neuroscience models is profitable for both parties - it is what lets us understand systems better and reveal new mechanisms. We have shown:
> 1) For neuroscientists: This  mathematical relationship we show helps us get a better understanding of hippocampal models. It also suggests a new mechanism for place cells that would not be possible without this mathematical relationship. It also sheds new light on the hippocampal role in complex sequence learning e.g. the hippocampal role in language has that seemed at odds with the traditional hippocampal role in spatial cognition and memory, but now can be unified by its relationship to transformers.
> 2) For ML researchers: We have uncovered new insights into position encodings in transformers. Before they were just thought of as conveying position in a line via sines and cosines. This was done to achieve better empirical results. Now we can understand them as reflecting the underlying structure of the problem (e.g. physical space in our case). This means that not every sequence should be given the same position encoding - as different sequences can take different trajectories, and so be at different positions.
>
> We have amended the paper, to make our main claims clearer, and to be clear about what our claims are not. We really appreciate your feedback here, as we had not realised that our work could be misinterpreted in this way - thanks!
>
> In particular in the main text we now say:
>
> “Note, we are not saying the brain is closely related to transformers because it learns the same neural representations, instead we are saying the relationship is close because we have shown a mathematical relationship between transformers and carefully formulated neuroscience models of the hippocampal formation. This relationship helps us get a better understanding of hippocampal models, it also suggests a new mechanism for place cells that would not be possible without this mathematical relationship, and finally it tells us something formal about position encodings in transformers.”

---

> > ### Comment · Reviewer_hKkg · 2021-11-29
> > **Thank you for the in-depth discussion!**
> >
> > Thank you for going in depth explaining the different aspects and probing various branches of the argument, this was very helpful for my understanding!
> >
> > Probably the main thing for me to realize here (and perhaps that is worth highlighting when presenting this work) is that the mathematical similarities between the models arise not only due to the modifications you make in Section 3, but that Transformer's overall mathematical model already has similarities with TEM model, that way it justifies why Transformer is of special interesting for this comparison.
> >
> > I still have some reservations, but one thing our discussion has demonstrated to me is that this work sparks an interesting discussion!
> > I am rising my score and hope to resume this at the conference :)

---

> > > ### Author Response · Authors · 2021-11-29
> > > **Many thanks!**
> > >
> > > No problem, and likewise - it has been a pleasure! Your comments have been really helpful to see where our paper was not clear, and have allowed us state our main claims in a much clearer way. The paper is much improved because of it.
> > >
> > > Hopefully we can talk more at the conference!

---

> ### Author Response · Authors · 2021-11-22
> **Response to Reviewer hKkg (2/4)**
>
> **REVIEWER COMMENT** “The emergent patterns resembling grid cell structure that are shown to emerge in this paper closely resemble the patterns that emerged in an RL agent in this work: https://deepmind.com/blog/article/grid-cells. In that work Transformers were not used, but the pattern has emerged nevertheless. This begs the question whether Transformers are indeed the architecture that encodes spatial representation closest to hippocampus, or whether any ML model, properly incentivized, will form such representations?“
>
> Many thanks for this comment - this raises a few points of interest here.
> 1) All models of grid cells that make next-step predictions use a RNN of some sort. The question is how they are trained. The DeepMind paper is trained to predict ground truth spatial representations of place cells (another paper from a similar time was trained on ground truth (x,y) locations). In both cases the modeller had to specify what the ground truth spatial representations look like, i.e. the experimenter had to solve the problem for the network already! This training is supervised. This is not how you or I learn. Instead, we learn by extracting regularities from the sensory world, and transfer/generalise this knowledge from domain to domain.
> 2) TEM was the first model to learn grid cell representations using sensory representations alone (unsupervised), and show that these grid cell representations can be transferred between environments to facilitate generalisation. It still uses a RNN for the grid cells, but now the grid cells also connect to sensory representations via hippocampal memories. A subsequent model (Uria et al., 2020) uses the same ideas to solve similar problems. Thus, all current evidence suggests that to learn and generalise grid cells representations via unsupervised learning requires a combination of a memory system and a RNN.
> 3) The relationship we show with transformers is not just about grid cells in entorhinal cortex, it is a relationship to the whole hippocampal-entorhinal system. This is exactly what we show - the memory part of the transformer is related to the hippocampal memory system in TEM. And the positional encodings are related to the entorhinal grid RNN system in TEM.
>
> We hope this is convincing that not just any ML system could learn and generalise grid cells via unsupervised learning. Instead, it needs 2 things - memories, and structured position representations.
>
> We have added in a clear section (‘related work’) relating this work to other models of grid cells, explaining how the transformer architecture fits in with existing work in the literature. We have not included the text here as it is several paragraphs long - please see the appendix!
>
> **REVIEWER COMMENT** “Authors mention in the abstract that "this result is no surprise since it is closely related to current hippocampal models from neuroscience" about the claim "transformers, when equipped with recurrent position encodings, replicate the precisely tuned spatial representations of the hippocampal formation". Could we have written the same paper by just replacing Transformers with something else that could mimic neuroscientific model of hippocampal grid cells? If yes, then Transformers have no special significance here and should not enter the discussion.“
>
> We hope that in the above comment, we have convinced you that to learn and generalise this knowledge via unsupervised learning, two things are necessary - a memory system, and a system for understanding ‘position’. This is exactly what TEM does. In this work, we have then shown that a model containing a memory system, and structured position representations (TEM) is closely related to transformers. **We are not saying the relationship is close because it produces the same representations, we are saying the relationship is close because we have shown a mathematical relationship between the two models.** This is not the case for most ML models since they do not have both RNN structured representations and a memory system and so is not mathematically relatable to current models of the hippocampal formation. A ML method that could do this are external memory systems (e.g. differentiable neural dictionaries) equipped with a RNN that indexes memories. But again, this is closely related to a transformer with recurrent positional embeddings.
>
> The defining feature of a transformer is self-attention via an inner product with queries and keys, and returning a weighted sum of values. This form of self-attention is what distinguishes a transformer from the more generally stated graph neural network. This type of attention is exactly the type of attention that TEM uses, as we show in our paper.
>
> We have added in a section in ‘related work’ that mentions these points. We have not included the text here as it is several paragraphs long - please see the appendix!

---

> > ### Comment · Reviewer_hKkg · 2021-11-29
> > **The main scientific contribution thus being...**
> >
> > Would it then be correct to say that the main contribution of your work was that you've demonstrated that in order to generalize to grid-cell like representations a model needs to have (a) memories, and (b) structured position representations? Should that be the title? :) Because if what you've shown is indeed the fact these two are the necessary components, then it has far-reaching implications (or it is a confirmation at this stage?) that our brain, in some form, should also have these mechanisms employed for this task.

---

> > > ### Author Response · Authors · 2021-11-29
> > > **Response to Reviewer hKkg**
> > >
> > > This is not our main result. The notion that solving these problems requires a memory systems and a position system existed in the TEM paper for example. Perhaps we calrified and emphasised it here, but we do not wish to overstate our claims or claim credit for ideas that may already exist. We agree that results is, however, far reaching and of deep importance to the brain.
> > >
> > > The main result of this paper is that transformers and models of the hippocampal formation can be seen as two sides of the same coin. This allows for neuroscientists to translate ideas from ML to neuroscience and visa versa. For example it offers a novel understanding of position encodings in transformers.
> > >
> > > Theory is often about showing that two disparate things are actually the same thing / related things but in two different lights. We have shown this for transformers and a model of the hippocampal formation.

---

> ### Author Response · Authors · 2021-11-22
> **Response to Reviewer hKkg (3/4)**
>
> **REVIEWER COMMENT** “Seeing how Transformers are very flexible models with high capacity, wouldn't it be the case that (with a little twist) one could fit them to become good representations of almost any phenomena? The theory that is put forward in this paper is that Transformers model representations that are close to those used by hippocampal formation. What kind of result would invalidate such a theory? Let's say that instead of Transformers we would use a black box method "X" that would tune its parameters in such a way, that they correlate well with hippocampal responses. Would that mean that "X" is good model of representations that are being used by hippocampus? Note that the more flexible "X" is and the more time and computational resources we have, the closer we can bring "X"'s activations to correlate with hippocampal responses. What would be a hypothetical empirical indication from your experiments that would lead you to conclude that "nah, Transformers are nothing like hippocampus"?“
>
> It is potentially the case that transformers can be trained on many different tasks, and that their representations can be compared to neural representations from specific brain areas (indeed this has been done in language areas - e.g. Schrimpf 2020). What is as yet unknown, is whether the various mechanistic models that already exist in neuroscience can be shown to be **mathematically** comparable to transformers. In our work **we are not saying the relationship is close because it produces the same representations, we are saying the relationship is close because we have shown a mathematical relationship between the two models**. Indeed, we agree with you that transformers replicating brain representations is not the same thing as transformers offering a mechanistic understanding of the computations in that brain region. In our work however, we have shown both directions: 1) Transformers can replicate brain representations in the Hippocampus and Entorhinal cortex, and 2) That existing mechanistic models of the Hippocampal formation are mathematically closely related to the transformer equations. This is a rare thing to happen between neuroscience and machine learning!
>
> It is also crucial to note that we did not train the transformers on grid representations. These representations emerged using only unsupervised learning of sensory prediction.
>
> **REVIEWER COMMENT** “Our brain did not learn in an environment with a 4-connected Eucledian structure. Wouldn't having this structure most assuredly lead to a representation that reflects the pattern of the structure? Let's say you would conduct the same experiment in an environment with a some kind of 10-connected star-shaped Eucledian structure -- wouldn't the patterns of representations formed by the Transformer would try to capture that star-shape?“
>
> It is the case that a 4-connected world is more likely to lead to square grid like representations (shown in Stachenfeld et al., 2017, Sorscher et al., 2019, Whittington et al., 2020). However, in Sorscher et al., they show that using a ReLu activation on the RNN breaks this symmetry and hexagonal grid cells hexagons are the favoured representation even in a 4-connected world.
>
> We choose to use 4-connected worlds, as that is what the TEM paper does. The TEM paper also considers 6-connected worlds as 6-connected worlds are more likely to get hexagonal grid cells even without the Relu non-linearity on the RNN. We additionally show, in the appendix, grid-like representation from TEM-t trained on 6-connected worlds.
>
> We note that for spatial graphs, we are limited to ‘regular tilings’ i.e. 3, 4 or 6-connected worlds. To path integrate we require an action to have the same ‘meaning’ everywhere, thus it is not possible to tile 2D with a 10-connected graph where each connection has an action that ‘means’ the same thing at every location.

---

> > ### Comment · Reviewer_hKkg · 2021-11-29
> > **Another way to ask**
> >
> > Here is another way to formulate my concern that perhaps will help me understand: Can we take some other model (not from the Transformer family), then make the modifications similar to the ones you do in Section 3 and obtain similar result about both the (a) emergent representations, and, more importantly (b) mathematical closeness.

---

> > > ### Author Response · Authors · 2021-11-29
> > > **Response to Reviewer hKkg**
> > >
> > > Thanks for this!
> > >
> > > The required components are a memory system and a system for understanding position. The TEM model turns out to be closely related to transformers, for example it uses inner products for indexing memories, which turns out to be closely analogous to the transformer self-attention. It may be possible to build other systems that use different types of self attention, but that would be one step away from mathematical closeness, even if they reproduce spatial repesentations.

---

> ### Author Response · Authors · 2021-11-22
> **Response to Reviewer hKkg (4/4)**
>
> **REVIEWER COMMENT** “page 3: "It now becomes interesting to see what representations are learned" -- taking into account the properties of the task and its spatial structure, what would we expect the representation that captures that structure to look like? Does it have to be a Transformer to capture that representation or could be captured, for example, by an autoencoder trained to find an efficient representation of the samples constituting the task?“
>
> We thank you for this question - it raises an interesting point.
>
> We are not saying that it is interesting that transformers learn grid cells for the sake of grid cells, instead we think it is interesting that they learn grid cells as it **says something about position encodings**. The mathematical parallel we have made from transformers to hippocampal models works in the other direction too - hippocampal models tell us something about transformers too! Namely that the positional encodings should express an abstract structuring of the task.
>
> In particular response to your point; solving this task looks like entering a novel environment, seeing a frog, then taking a north, east, and south action. Then being asked what comes after tasking a west, and correctly predicting a frog. An autoencoder could not solve this task since it learns slow statistical structures between sensory observations over many training examples. Thus, it would be clueless to the entirely new configuration of sensory observations in the new environment. To solve this task, you need 1) A position understanding to know where you are and predict the consequence of actions and 2) A memory to remember what you saw at each position. An autoencoder cannot serve as either component.
>
> It is true, though, that training an autoencoder on place cell representations will get you grid cell representations (Dordek et al., 2016). But this does not actually solve any task in the real world. The brain faces problems like predicting what will come next, or the consequences of actions. Thus, the brain needs to store models of the world, and not just correlations (like an autoencoder does). Both TEM and TEM-t are trained using sensory observations alone, just like the brain.
>
> We now discuss this in the ‘related work’ section in the appendix.
>
> **REVIEWER COMMENT** “page 4: "Hopfield network (Hopfield, 1982) which have recently been shown to be closely related to transformers" -- Could you please elaborate what is meant by being related and how this relationship pertains to the claim that "a transformer with recurrent positional encodings is closely related to current neuroscience models of the hippocampus and surrounding cortex"?“
>
> In Ramsauer et al., 2020, and Krotov & Hopfield 2020, it is shown that the transformer update is closely related to the memory retrieval process of a Hopfield network (see discussion with reviewer Rqx3). In particular when optimising the Hopfield energy, one recovers the transformer update equation. This requires a particular form of the hopfield energy - one where the hopfield energy is of the form -ln(sum_k exp(...)), whereas the classical Hopfield network equations are recovered if the Hopfield energy is of the form (...)^2.
>
> This is, however, just the memory network side of things. Current models of the hippocampal formation consist of a Hopfield network for memories and a RNN for ‘position’ understanding and indexing memories. In this work we show these two components can be seen as a transformer with recurrent positional embeddings. This offers new insights for neuroscientists, and new interpretations of what positional encodings are.
>
> **REVIEWER COMMENT** “page 2: typo "Crucially, such rules generalises"”
>
> Fixed!

---

### Official Review · Reviewer_RQx3 · 2021-11-03

**Correctness:** 3
**Technical Novelty And Significance:** 3
**Empirical Novelty And Significance:** 3
**Recommendation:** 8
**Confidence:** 4

**Main Review:**

This paper nicely connects several ideas discussed recently in the AI and comp. neuroscience communities, such as transformers, Hopfield Networks, memory systems, place cells in the hippocampus, grid cells, etc. I want to give a couple of pointers to the literature that might make these connections more transparent and mathematically more precise.

1. The left equation in (Eq 11) seems to be closer related to the class of models that have power activation function (see for example equation 10 in Dense Associative Memory for Pattern Recognition by Krotov & Hopfield 2016 https://arxiv.org/abs/1606.01164, and assume that f is linear, or more generally a power function) than to models with f=softmax. In the classification scheme of Krotov & Hopfield 2020, the former is a class of models that they call models A, while the latter is the class of models that they call models B. The same applies to (Eq 8).

2. The network motif with two kinds of feature neurons, similar to the one shown in Fig.5 B is commonly used in models of associative memory in the hetero-associative setting. When there are two kinds of features that need to be associated with each other in one memory. For instance, this idea was used in Krotov & Hopfield 2016 for associating pixel intensities of images with the labels of those images, see for example Figure 1A. The pixel intensities are presented to the network and hold clamped, while the label part of the feature neurons are updated to compute the desired output.

3. The sentence “It has been shown that the use of a softmax corresponds to a variant of the Hopfield energy (Demircigil et al., 2017) where the number of memories is untethered from dimensionality of the attractor and therefore are potentially unbounded.” on page 8 is not entirely correct. First, the model by Demircigil and coauthors has a bounded capacity, which is at most $2^{(N/2)}$, see their theorem 1.3 in https://arxiv.org/abs/1702.01929. Second, the activation function in that model is f(x) = exp(x), rather than the softmax. I understand that the confusion comes from the derivation given in Ramsauer et al., who start with the model with f(x)=exp(x), then take a logarithm of the energy function for that model, and add an extra term to enforce the lower bound on the energy. The problem with this logic is that the addition of the quadratic term changes the mathematical properties of the model. For this reason, the model with the softmax activation, and the model with f=exp(x) are two mathematically different models.

I also wish the paper had some kind of quantitative metrics of how closely the learned representations resemble place cells. Right now the similarity is only qualitative, e.g. Figure 5D. If the evaluation was somewhat more quantitative the authors might be able to ask interesting questions. For example how does the choice of the activation function (softmax, power, exponent) change the similarity of the learned representations to biological place cells? Maybe some activation functions model the biological system better than others?


**Summary Of The Paper:**

This paper demonstrates emergence of grid and place cells in transformer architectures with positional encodings. Transformers are related to TEM models and Hopfield networks in computational neuroscience. A neurobiological model of TEM-transformers is presented, which contains feature neurons and memory neurons. It is shown that memory neurons resemble place cells in the hippocampus. Remapping phenomenon is also discussed.

**Summary Of The Review:**

In general, I like the scope of the paper, and I am inclined to vote accept provided that the authors can address the comments above.

---

> ### Author Response · Authors · 2021-11-22
> **Response to Reviewer RQx3 (1/3)**
>
> **REVIEWER COMMENT** “This paper nicely connects several ideas discussed recently in the AI and comp. neuroscience communities, such as transformers, Hopfield Networks, memory systems, place cells in the hippocampus, grid cells, etc. I want to give a couple of pointers to the literature that might make these connections more transparent and mathematically more precise.”
>
> Many thanks - much appreciated! We are very pleased that you liked the connection between AI models and models from neuroscience. We appreciate your comments that we were not as precise as we could have been - we agree! Many thanks for pointing this out to us, the paper is better for it!
>
> **REVIEWER COMMENT** “The left equation in (Eq 11) seems to be closer related to the class of models that have power activation function (see for example equation 10 in Dense Associative Memory for Pattern Recognition by Krotov & Hopfield 2016 https://arxiv.org/abs/1606.01164, and assume that f is linear, or more generally a power function) than to models with f=softmax. In the classification scheme of Krotov & Hopfield 2020, the former is a class of models that they call models A, while the latter is the class of models that they call models B. The same applies to (Eq 8).”
>
> You’re exactly right - thanks for pointing this out to us! Eq 11 is the case where the power activation function on the Hopfield Energy is of order 2. Indeed, in Krotov & Hopfield 2016, and later Demircigil et al., 2017, they show that higher powers have improved memory capacity scaling properties - in essence because now memories are not just bound via pairwise interactions between elements of the vector, but now 3-wise interactions and higher.
>
> We have now clarified the relationship of our equations to Krotov & Hopfield 2020 type B models. We say
>
> "As a note, the particular relationship of our model to the model of Krotov & Hopfield 2020, is what they refer to as a 'type B' model. These are models with contrastive normalisation on the memory neurons (via a softmax in our case), as opposed to `type B' models which have a power activation function on the memory neurons. TEM (left hand side of Equation 11) corresponds to a linear activation function on the memory neurons, and is directly analogous to the original Hopfield energy.”
>
> **REVIEWER COMMENT** “The network motif with two kinds of feature neurons, similar to the one shown in Fig.5 B is commonly used in models of associative memory in the hetero-associative setting. When there are two kinds of features that need to be associated with each other in one memory. For instance, this idea was used in Krotov & Hopfield 2016 for associating pixel intensities of images with the labels of those images, see for example Figure 1A. The pixel intensities are presented to the network and hold clamped, while the label part of the feature neurons are updated to compute the desired output.”
>
> Interesting, thank you for the reference! We now mention this and refer to Krotov & Hopfield 2016, and show that our model is similar but now where one of the representations is structured (via learning) according to the underlying task structure. We say:
>
> “Secondly, the notion that there are two types of feature neurons that can be bound together in the same memory, was explored in Krotov & Hopfield 2016 where pixel intensities were associated with labels of those images. In TEM-t, one of the feature vectors, g, is learned via a RNN and structures itself according to the underlying task structure.”

---

> ### Author Response · Authors · 2021-11-22
> **Response to Reviewer RQx3 (2/3)**
>
> **REVIEWER COMMENT** “The sentence “It has been shown that the use of a softmax corresponds to a variant of the Hopfield energy (Demircigil et al., 2017) where the number of memories is untethered from dimensionality of the attractor and therefore are potentially unbounded.” on page 8 is not entirely correct. First, the model by Demircigil and coauthors has a bounded capacity, which is at most  2(N/2), see their theorem 1.3 in https://arxiv.org/abs/1702.01929. Second, the activation function in that model is f(x) = exp(x), rather than the softmax. I understand that the confusion comes from the derivation given in Ramsauer et al., who start with the model with f(x)=exp(x), then take a logarithm of the energy function for that model, and add an extra term to enforce the lower bound on the energy. The problem with this logic is that the addition of the quadratic term changes the mathematical properties of the model. For this reason, the model with the softmax activation, and the model with f=exp(x) are two mathematically different models.”
>
> You’re exactly right that the number of memories is not unbounded but rather follows the scaling law you describe. Many thanks for pointing this out - we have amended the text accordingly. We change the paragraph to say:
>
> “As an additional aside, we note that Krotov & Hopfield 2020 architectures does not solve the scaling problem of conventional Hopfield networks; the number of memories that the original Hopfield networks could store scaled linearly with the dimensionality of the recurrent attractor network (Amit et al 1985). While recent analytical work has shown with exponential power activation functions, the number of memories that can be stored to scale as \(2^{\frac{N}{2}}\), where N is the dimensionality of the feature neurons (Demircigil et al 2017). This is a considerably more favourable scaling. However, unfortunately the architecture from Krotov & Hopfield 2020 instead tethers it to the number of memory neurons, so the number of memories is still linear with the number of neurons! We note that mathematically derived scaling law was for an exponential activation function, not with a softmax as we use here.”
>
> You’re also right that f=exp() and f=softmax() aren’t the same and so the mathematical properties cannot be directly transferred. We have amended the statement to say that the softmax is closely related to the work of Demircigil, but the mathematical statement of the scaling law does not necessarily transfer to our situation. At the end of the corresponding paragraph (shown above), we say
>
> “We note that mathematically derived scaling law was for an exponential activation function, not with a softmax as we use here.”

---

> ### Author Response · Authors · 2021-11-22
> **Response to Reviewer RQx3 (3/3)**
>
> **REVIEWER COMMENT** “I also wish the paper had some kind of quantitative metrics of how closely the learned representations resemble place cells. Right now the similarity is only qualitative, e.g. Figure 5D. If the evaluation was somewhat more quantitative the authors might be able to ask interesting questions. For example how does the choice of the activation function (softmax, power, exponent) change the similarity of the learned representations to biological place cells? Maybe some activation functions model the biological system better than others?”
>
> **Grid cells**: We have now included the ‘grid-ness’ score - a metric from neuroscience for determining how grid-like a grid cell is. We have also added a figure in the appendix which shows all ‘positional RNN’ cells learned from a particular model run (both with a linear activation on the RNN, and a ReLu activation on the RNN. These cells are ordered according to their gird score.
>
> **Place cells**: We are not aware, unfortunately, of any metrics from neuroscience to determine how place-like a place cell is (nor are we aware of a metric for band cells). We, however, develop a new metric to assess how place like the firing of each cell is. In particular, we look at the connected components of the firing rate map, and our metric is the ratio of ‘firing mass’ in the largest connected component versus all connected components. This metric is 1 if all the firing is in a single component, and it is lower if the firing is spread between components. We have also added a figure in the appendix which shows all TEM-t memory ‘place’ cells, sorted by our new place cell metric.
>
> (We do not claim this metric to be perfect for place cells in general, but it suits our situation well. In the appendix, we show that this metric is high for TEM-t ‘place’ cells and lower for TEM-t grid cells. I.e. the metric appropriately separates out model ‘place’ and ‘grid’ cells.)
>
> It is an interesting question how the choice of activation function affects the representations. Default TEM just uses a linear activation function, whereas TEM-t uses a softmax activation. As you say above, power and exponential activations allow for greater memory capacity. However, we suspect place like representations are possible even without a softmax activation; since memories are ‘attended’ to via a dot product, it would be possible to have sparsely activated memory neurons without using a softmax - for example all memories just need to have dot product ~1 for the correct memory, and dot product ~0 for incorrect memories. For this, a softmax is not required, but likely helps stabilise learning via its normalising effect. We agree that it would be interesting to investigate this experimentally, though unfortunately we have not had enough time to run these experiments. Nevertheless, we include text in the appendix which discusses these issues. We will endeavour to run these experiments in due course. In the appendix we say:
>
> “Place-like representations without a softmax: Here we chose a softmax activation function on the memory neurons to make the relationship between TEM and transformers exact. However was this choice necessary to observe place cells in the memory neurons? While we have not examined this experimentally, we suspect place-like representations would emerge for a variety of activations, since memories must still be sparsely activated for correct prediction. We leave this for future work to verify, but note again that power and exponential activations have been shown to have a much greater memory capacity (Krotov & Hopfield 2016, Demiircigil 2017) than traditional Hopfield nets with linear activation."

---

> > ### Comment · Reviewer_RQx3 · 2021-11-27
> > **Thanks for the revisions, I have increased my score.**
> >
> > Thank you for the revisions. I think the paper has improved and became more accurate. I have increased my initial score to full accept. One little comment: there is a typo on page 8, in the sentence “These are models with contrastive normalisation on the memory neurons (via a softmax in our case), as opposed to ‘type B’ models which have a power activation function on the memory neurons.” It should be ‘type A’, not ‘type B’. Type A models use neuron-wise activation (e.g. power function ), type B models use contrastive normalization (e.g. softmax).

---

> > > ### Author Response · Authors · 2021-11-28
> > > **Many thanks!**
> > >
> > > Many thanks - we are very grateful that you have raised the initial score :). Overall, we have really appreciated your comments, and they have greatly improved the paper
> > >
> > > When the paper is published (on iclr or otherwise), we will fix the typo.
> > >
> > > Many thanks once again!

---

> ### Author Response · Authors · 2021-11-22
> **Delayed thanks for the review!**
>
> Many thanks for your review! We are very grateful for your kind apprecation of our work, and your critiscims have been taken onboard and have greatly impreoved the manuscript. We hope that we have addressed your comments in our response below!

---

### Official Review · Reviewer_K1VL · 2021-11-03

**Correctness:** 4
**Technical Novelty And Significance:** 3
**Empirical Novelty And Significance:** 3
**Recommendation:** 8
**Confidence:** 4

**Main Review:**

Major issues:
- Many papers have reported finding learned representations that resemble place cells and some have found grid cells as well. It would be quite helpful to contextualize the findings in this manuscript with others, as well as more rigorously evaluate the degree of evidence for these types of response properties, by: (1) defining a metric of place/grid cell-ness, and then quantifying the extent to which a given model unit resembles a place cell or grid cell, and then (2) quantifying how common these representations are across all model units, rather than simply showing one or two example units (and ideally comparing these numbers to estimates of the prevalence of such units in the relevant [biological] neural populations). And then (3) showing the top K units that score on this rather than one or two (ideally k=50 or 100 and then is just a big appendix figure). If the authors also wish to discuss band cells then they should do this for that type as well. Lastly, an obvious gap is that the authors don't mention boundary cells -- do they find any units that mimic those?

- I found myself wanting a full end-to-end diagram of both models, but particularly the TEM, as I am (and I imagine the ICLR audience more broadly is) less familiar with it. Obviously, space is tight, but including a full schematic of each model would be  particularly helpful. If need be, this could be in the appendix.

- Further clarity on what the task exactly is. The authors briefly discuss it briefly in a few paragraphs, but the manuscript would benefit from more details on the exact nature of the task -- e.g., example diagrams of the environments, and perhaps even examples of performance over training. Since space is tight, this could just be in the appendix.


Minor issues:
- What do the authors mean that TEM-t both has better sample efficiency and has reduced training time? Naively, one would imagine these things are largely equivalent statements, but I assume they mean something more here.
- Typos:
  - Pg 5: "via via Hebbian learning" --> "via Hebbian learning"
  - Pg 6: "TEM can tackle much larger problems" --> "TEM-t can tackle..."
  - Pg 7: "relation ship" --> "relationship"


**Summary Of The Paper:**

This well-written and clear paper clarifies the relationship between transformers and a recent exciting model of the medial temporal lobe in neuroscience. The authors find that a transformer with recurrent positional encodings trained on a spatial navigation task ends up learning representations that resemble some classic findings from the medial temporal lobe in the neuroscience (e.g., grid and place cells). This finding alone is not that surprising -- as the authors note, a variety of groups have previously found such representations are learned in navigation tasks with other models (particularly place cells). The major contribution of this paper is that the authors then demonstrate the connection between transformers and a contemporary model of the hippocampal formation, the Tolman-Eichenbaum Machine (TEM). Specifically, the memory retrieval process in the TEM resembles self-attention and the path integration representations are comparable to a transformer's learned position encodings. I believe that by offering a novel perspective on transformers and models of biological memory systems, this work may lead to fruitful future work both in machine learning and neuroscience.


**Summary Of The Review:**

By offering a novel perspective on transformers and models of biological memory systems, this work may lead to fruitful future work both in machine learning and neuroscience.

---

> ### Author Response · Authors · 2021-11-22
> **Response to reviewer K1VL**
>
> Many thanks for your reviews - we are very happy that you liked the paper! We appreciate your comments that the paper could do with quantifying the representations, and more explanations and figures of the models - we have amended the paper to include these!
>
> **REVIEWER COMMENT** “Many papers have reported finding learned representations that resemble place cells and some have found grid cells as well. It would be quite helpful to contextualize the findings in this manuscript with others, as well as more rigorously evaluate the degree of evidence for these types of response properties, by: (1) defining a metric of place/grid cell-ness, and then quantifying the extent to which a given model unit resembles a place cell or grid cell, and then (2) quantifying how common these representations are across all model units, rather than simply showing one or two example units (and ideally comparing these numbers to estimates of the prevalence of such units in the relevant [biological] neural populations). And then (3) showing the top K units that score on this rather than one or two (ideally k=50 or 100 and then is just a big appendix figure). If the authors also wish to discuss band cells then they should do this for that type as well. Lastly, an obvious gap is that the authors don't mention boundary cells -- do they find any units that mimic those?“
>
> **Grid cells**: We have now included the ‘grid-ness’ score - a metric from neuroscience for determining how grid-like a grid cell is. We have also added a figure in the appendix which shows all ‘positional RNN’ cells learned from a particular model run (both with a linear activation on the RNN, and a ReLu activation on the RNN. These cells are ordered according to their gird score.
>
> **Place cells**: We are not aware, unfortunately, of any metrics from neuroscience to determine how place-like a place cell is (nor are we aware of a metric for band cells). We, however, develop a new metric to assess how place like the firing of each cell is. In particular, we look at the connected components of the firing rate map, and our metric is the ratio of ‘firing mass’ in the largest connected component versus all connected components. This metric is 1 if all the firing is in a single component, and it is lower if the firing is spread between components. We have also added a figure in the appendix which shows all TEM-t memory ‘place’ cells, sorted by our new place cell metric.
>
> (We do not claim this metric to be perfect for place cells in general, but it suits our situation well. In the appendix, we show that this metric is high for TEM-t ‘place’ cells and lower for TEM-t grid cells. I.e. the metric appropriately separates out model ‘place’ and ‘grid’ cells.)
>
> **Border cells**: We were also interested that we did not observe there, since TEM did. A likely reason is that the TEM grid cell RNN can get lost easily (since path integration is inherently noisy and that the location correction from the attractor network is not brilliant as TEM is not great at forming and retrieving memories). Thus using border cells - which anchor at the border is a way of aligning representations and not getting lost so easily. We have this problem less since our memory network is very good and so it is easier to self-localise.
>
> **REVIEWER COMMENT** “I found myself wanting a full end-to-end diagram of both models, but particularly the TEM, as I am (and I imagine the ICLR audience more broadly is) less familiar with it. Obviously, space is tight, but including a full schematic of each model would be particularly helpful. If need be, this could be in the appendix.“
>
> Many thanks! We have included a more detailed description of TEM in the appendix, along with a companion figure. We do not include the text here as it is several paragraphs - please see the appendix for new changes!

---

> > ### Comment · Reviewer_K1VL · 2021-11-22
> > **What proportion of your units are grid cell-like or place cell-like?**
> >
> > Thanks for the thorough responses! The one thing I don't see in the updated manuscript that I think is critical: A histogram for grid score for each unit and a histogram of place score for each unit. I think it's pretty important to have a sense of how representative these units are in your whole population.

---

> > > ### Author Response · Authors · 2021-11-22
> > > **Grid score distribution added for all RNN cells. Place score distribution added for all memory cells.**
> > >
> > > Thanks for the quick response!
> > >
> > > We've added the distribution of grid scores for all RNN neurons in the population next to the ratemaps (Figure 9 and 10).
> > >
> > > We've also done the same for the new place score metric for all memory neurons (Figure 11).

---

> ### Author Response · Authors · 2021-11-22
> **Response to reviewer K1VL (2/2)**
>
> (cont)
>
> **REVIEWER COMMENT** “Further clarity on what the task exactly is. The authors briefly discuss it briefly in a few paragraphs, but the manuscript would benefit from more details on the exact nature of the task -- e.g., example diagrams of the environments, and perhaps even examples of performance over training. Since space is tight, this could just be in the appendix.“
>
> Many thanks! We have included an extended task description, along a figure, in the appendix. We do not include the text here as it is several paragraphs long - please see the appendix.
>
> **REVIEWER COMMENT** “What do the authors mean that TEM-t both has better sample efficiency and has reduced training time? Naively, one would imagine these things are largely equivalent statements, but I assume they mean something more here.“
>
> Apologies for the confusing wording. By sample efficiency we mean it takes fewer data examples to reach good performance. By reduced training time: We had intended for reduced training time to mean that each batch is processed with less time than TEM. While this is true, we realise now it is confusing and so we remove the statement. Now we say:
>
> “In particular, 1) Sample efficiency is increased - TEM-t requires many fewer data samples than TEM, and thus training time is reduced.”
>
> **REVIEWER COMMENT** Pg 5: "via via Hebbian learning" --> "via Hebbian learning"
> Pg 6: "TEM can tackle much larger problems" --> "TEM-t can tackle..."
> Pg 7: "relation ship" --> "relationship"
>
> Fixed!

---

> > ### Comment · Reviewer_K1VL · 2021-11-29
> > **Thanks for thorough engagement with reviewers**
> >
> > I have read through the reviews and responses. I think the paper is much improved through this process -- the authors engaged earnestly and thoroughly. I am standing by my score of 8.

---

> > > ### Author Response · Authors · 2021-11-29
> > > **Many thanks!**
> > >
> > > Many thanks for this! We really appreciate your time and effort in helping us make this paper better!

---

### Decision · Program_Chairs · 2022-01-20

**Decision:**

Accept (Poster)

**Comment:**

This paper received 4 unanimous accept (including 1 marginal accept). This well-written and clear paper clarifies the relationship between transformers and a recent exciting model of the medial temporal lobe in neuroscience. There was some clarifications requested by the reviewers that were addressed during the revision. This paper will make a great computational neuroscience contribution to this year ICLR!